# Crystallographic analysis of temperate ice on Rhonegletscher, Swiss Alps

Sebastian Hellmann[1,2], Johanna Kerch[3,*], Ilka Weikusat[3,4], Andreas Bauder[1], Melchior Grab[1,2], Guillaume Jouvet[5,6], Margit Schwikowski[7], and Hansruedi Maurer[2]

[1]Laboratory of Hydraulics, Hydrology and Glaciology (VAW), ETH Zurich, Zurich, Switzerland
[2]Institute of Geophysics, ETH Zurich, Zurich, Switzerland
[3]Alfred-Wegener-Institut Helmholtz-Zentrum für Polar- und Meeresforschung, Bremerhaven, Germany
[4]Department of Geosciences, Eberhard Karls University, Tübingen, Germany
[5]Department of Geography, University of Zurich, Zurich, Switzerland
[6]Autonomous Systems Laboratory, ETH Zurich, Zurich, Switzerland
[7]Paul Scherrer Institute, Villingen, Switzerland
[*]now at: GZG Computational Geoscience, Georg-August University, Göttingen, Germany

**Correspondence:** Sebastian Hellmann (sebastian.hellmann@erdw.ethz.ch)

**Abstract.** The crystal orientation fabrics (COF) analysis provide information about the c-axis orientation of ice grains and the associated anisotropy and microstructural informtion about deformtation and recrystallisation processes within the glacier. These information can be used to introduce modules that fully describe the microstructural anisotropy or at least direction-dependent enhancement factors for glacier modelling. The COF was studied at an ice core that was obtained from the temperate Rhonegletscher, located in the Central Swiss Alps. Seven samples, extracted at depths between 2 and 79 m, were analysed with an automatic fabric analyser. The COF analysis revealed conspicuous four-maxima patterns of the c-axis orientations at all depths. Additional data, such as microstructural images, produced during the ice sample preparation process, were considered to interpret these patterns. Furthermore, repeated high-precision Global Navigation Satellite System (GNSS) surveying allowed the local glacier flow direction to be determined. The relative movements of the individual surveying points indicated longitudinal compressive stresses parallel to the glacier flow. Finally, numerical modelling of the ice flow permitted estimation of the local stress distribution. An integrated analysis of all the data sets provided indications and suggestions for the development of the four-maxima patterns. The centroid of the four-maxima patterns of the individual core samples and the coinciding maximum eigenvector approximately align with the compressive stress directions obtained from numerical modelling with an exception for the deepest sample. The clustering of the c-axes in four maxima surrounding the predominant compressive stress direction is most likely the result of a fast migration recrystallisation. This interpretation is supported by air bubble analysis of Large-Area Scanning Macroscope (LASM) images. Our results indicate that COF studies, which were so far predominantly performed at cold ice samples from the polar regions, can also provide valuable insights on the stress and strain rate distribution within temperate glaciers.

# 1 Introduction

Since the second half of the last century, ice cores have been regarded as extremely valuable archives for reconstructing the climate history of the past hundred-thousands of years (Robin et al., 1977; Petit et al., 1999; Thompson et al., 2002). For example, correlations between ice accumulation, isotopes and dust content have been established, but the deformation of ice layers complicates dating and interpretation of climate records (Jansen et al., 2016). Microstructural analyses have been used to overcome these issues (Faria et al., 2010). In addition, microstructural investigations have also been conducted to reconstruct the ice flow of ice sheets in Greenland and Antarctica as well as in glaciated mountain areas (Russell-Head and Budd, 1979; Alley, 1992; Azuma, 1994). For those investigations, the focus has been on the crystallographic orientation of the ice grains. The stresses and strain rates occurring within the ice mass not only cause glacier flow, but also induce the development of a characteristic COF and microstructural anisotropy (Gow and Williamson, 1976; Herron and Langway, 1982; Alley et al., 1995, 1997) and summarised in Faria et al. (2014a).

During the past decades, COF and texture have been investigated intensively on polar deep ice cores to understand the microstructure of polycrystalline ice in the context of its deformation history (Hooke, 1973; Gow and Williamson, 1976; Thorsteinsson et al., 1997; Patrick et al., 2003; Gow and Meese, 2007; Pettit et al., 2007; Montagnat et al., 2014; Pettit et al., 2011; Weikusat et al., 2017). A historical summary of these projects can be found in Faria et al. (2014a). For the selected ice core drilling spots on domes and ridges, vertical compression and horizontal extension within the ice mass have been found to be the dominant driving stress for ice deformation. In contrast, for ice samples from temperate glaciers, the deformation is dominated by a series of interfering and changing compressional, extensional, and shear stress conditions along the valley. Together with a diagenesis, burial, basal sliding, and potentially partial melting these stress conditions results in a much more complex deformation history (Hambrey and Milnes, 1977). This requires more extensive analyses of COF.

The ice of temperate glaciers is comparable with a metamorphic rock close to its melting point (Hambrey and Milnes, 1977) that has been exposed to a long series of deformation processes along the valley. This deformation is caused by various shear and compressional stresses that have been applied to the ice. These stress regimes produce heterogeneously distributed dislocations, which cause dynamic recrystallisation by rearrangement of these dislocations and by internal strain energy reduction. The resulting recrystallisation processes and the interplay between deformation and recrystallisation in the ice take place even faster as the temperature gets closer to the pressure melting point (Alley, 1988; Weikusat et al., 2009a). As a result, the adaption of the ice crystal structure to new stress conditions is expected to be faster (e.g. Kamb, 1972; Duval, 1979). Additionally, the higher temperatures provide more thermal energy and allow a faster grain growth (Azuma et al., 2012), leading to an interplay between stress and temperature regime (Alley, 1988; Faria et al., 2014b). Therefore, large differences can be observed between cold and temperate ice. One of the most apparent differences is the grain size, which has been found to be a few centimetres in temperate ice (Rigsby, 1960), whereas samples from polar ice usually show grains with a diameter of a few millimetres, except in the deepest parts, where temperatures rise close to the pressure melting point (e.g. Gow and Williamson, 1976; Thwaites et al., 1984; Kuiper et al., 2019).

First crystallographic investigations have been performed on temperate glaciers already in the 1950's to 1980's, including the

detailed investigations of Kamb (1959) and Rigsby (1960), and later extended by Budd (1972), Hambrey and Milnes (1977), Hooke and Hudleston (1978), and Hambrey et al. (1980). A potential problem of temperate glacier crystal analysis is the large grain size and thus limited amount of grains that can be analysed for each sample. This may be the reason, why a surprisingly low number of papers has been published on crystal structure of temperate glaciers (e.g. Tison and Hubbard, 2000) during the past years. Furthermore, the majority of the earlier studies mainly analysed samples from the uppermost few meters. To date, ice core drilling and preparation of thin sections is still a time-consuming process. Only a few discrete measurements are possible within a reasonable amount of time. Nonetheless, the technique for analysing COF has developed extensively, for example, by using image analysis software and powerful computing resources (Wilson et al., 2003; Peternell et al., 2009; Wilson and Peternell, 2011; Eichler, 2013).

In this study, we analyse ice core samples from a temperate alpine glacier. We describe and compare our findings with previous studies and provide a hypothesis for the resulting COF in terms of given stress and temperature conditions. We analyse the stress regime in the vicinity of the ice core, using additional borehole measurements and discuss recrystallisation processes and grain growth in temperate ice. For selected examples we take a closer look at the development of new ice crystals under the current stress regime of the glacier. The microstructural results of this study serve as a basis for geophysical experiments on ice core samples and they can also be compared with results from larger scale geophysical experiments.

## 2  Field Site and Data Acquisition

The field work was carried out on Rhonegletscher, located in the Central Swiss Alps (Fig. 1). This glacier currently covers an area of about $15.5\,\mathrm{km}^2$ and is flowing in a southern direction from 3600 m a.s.l. down to 2200 m a.s.l. It is a medium-sized valley glacier, easily accessible, and therefore investigations had been carried out already in the last two centuries and continuously since 2006 (Bauder, 2018).

In August 2017, we drilled an ice core in the ablation area of the glacier (Fig. 1), approximately 500 m north of its current terminus. Here, the ice was flowing with an average surface velocity of $16.2\,\mathrm{m\,a}^{-1}$ in the season 2017/18 according to GNSS measurements. This location was selected, because the glacier surface forms a relatively even plateau with only 5 m elevation change over a distance of 40 m and is free of crevasses. Further up-glacier there is a steep and crevassed area. An analysis of the bedrock with ground-penetrating radar measurements also confirmed a transition from a steep to a more flat zone of the valley (Church et al., 2018) at the ice core location.

As the ice is just at the pressure melting point, we used a thermal drilling technique (Schwikowski et al., 2014). Although hot-water drillings, performed in the vicinity of the ice core location, showed a mean ice thickness of 110 m, we stopped drilling at 80 m, when hitting some gravel. This gravel blocked the cutter head. We retrieved an 80 m long ice core, with a gap between 46 and 50 m due to technical issues.

Due to the thermal drilling technique, which did not apply a rotational force onto the ice core segments, an oriented retrieval of the segments was possible. A freshly drilled segment was manually connected to the previous one, which worked out well for most of the segments. Additional measurements of the Earth's magnetic field, while drilling, could be used in some cases

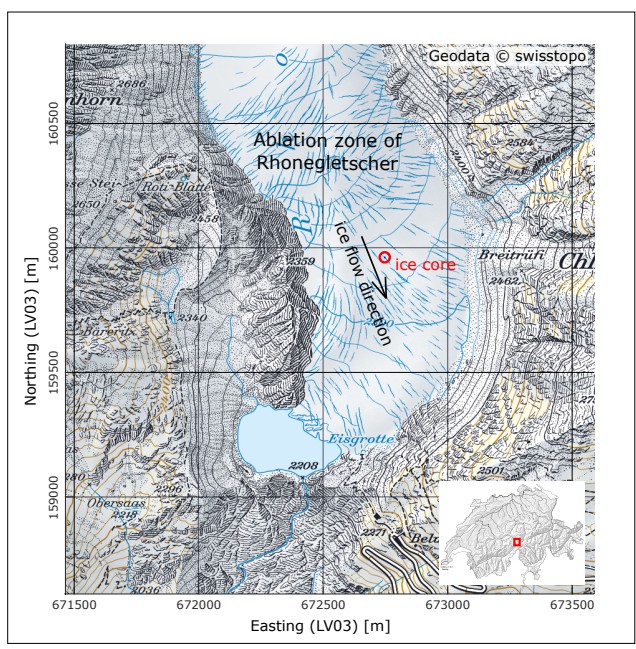

**Figure 1.** Rhonegletscher ablation area, ice core position indicated in red, ice flow direction at ice core location shown by black arrow. Map source: Swiss Federal Office of Topography.

to reconstruct the orientation within a range of $\pm 10°$ when matching of neighbouring segments was not possible.

To complement the results of the ice core analysis, we made use of an array of hot-water-drilled boreholes surrounding the location of the ice core retrieval (Fig. 2). The locations of the borehole collars were surveyed repeatedly using high-precision GNSS measurements (Fig. 2a). The displacements of the borehole collars indicate a south-eastern flow direction with an az-

90 imuth of about $155° \pm 10°$. We further derived the surface velocities at each borehole location (Fig. 2b). The south-eastern boreholes (BH04 to BH07) show significantly smaller displacements, compared with the boreholes located in the north-western part of the array (BH01 and BH10 to BH12). This indicates compression of the ice in this region, which is expected to lead to larger longitudinal strain rates (convention: compressional strain rates = negative values). We calculated the surface strain rate components $\dot{\epsilon}_{xx}$, $\dot{\epsilon}_{xy}$, and $\dot{\epsilon}_{yy}$ (Fig. 3) from these velocities. These strain rates and velocities serve as constraints for the ice

flow modelling.

In the following sections, we refer to $x$ as the component in glacier flow direction (i.e. $\approx 155°$ from North, cf. Fig. 1), $y$ is the component perpendicular to the glacier flow ($\approx 65°$ from North), and $z$ is oriented vertically upwards.

## 3 Ice flow modelling

To support these observations at the surface, the internal glacier dynamics was investigated by the means of a three-dimensional

ice flow model that already existed from previous studies (Jouvet et al., 2009; Morgenthaler, 2019). An updated bedrock model

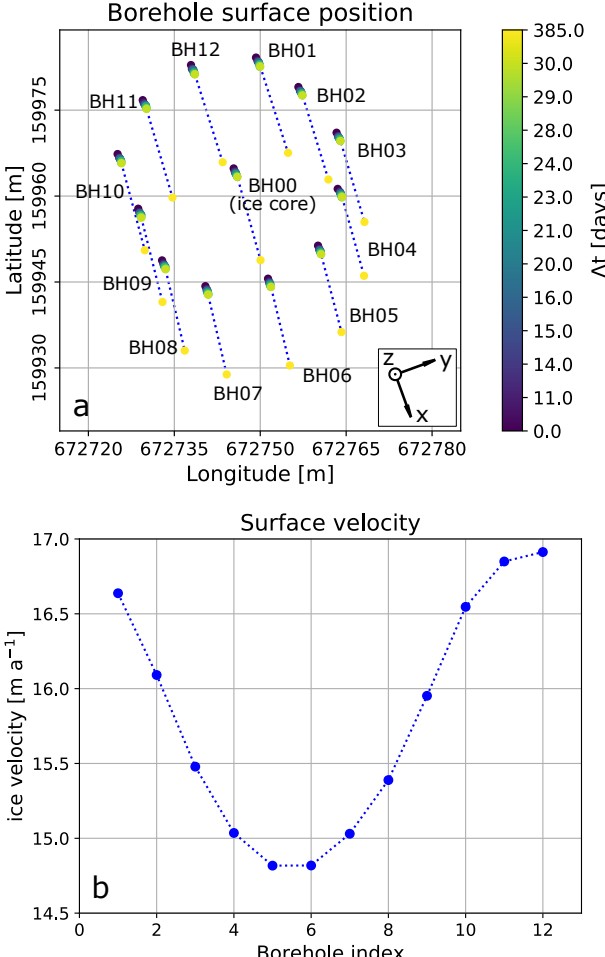

**Figure 2.** Analysis of ice flow direction and velocity for each borehole, (a) the displacement of the borehole surface points measured by GNSS since drilling, inset depicts the glacier flow coordinate system, (b) the absolute surface velocity [$\mathrm{m\,a^{-1}}$] in the vicinity of each borehole.

and surface topographic information were used to determine the actual ice thickness of the glacier and to constrain the model. The bedrock model was obtained from GNSS measurements (Church et al., 2018) and from a Swiss-wide glacier inventory that is currently being updated (Farinotti et al., 2009; Grab et al., 2018). With the given information, we simulated the ice flow of Rhonegletscher using the Elmer/Ice modelling code (Gagliardini et al., 2013), which solves the full Stokes equations based
on Glen's flow law (Glen, 1955) for the ice rheology, namely,

$$\dot{\epsilon}_{ij} = A(T)\,\tau^{n-1}\,\tau_{ij}, \tag{1}$$

where $\dot{\epsilon}_{ij}$ is the strain rate, $\tau_{ij}$ is the deviatoric stress, and $\tau$ is the second invariant of the deviatoric stress tensor. The creep exponent $n$ was set to $n = 3$ as typical value for valley glaciers (Budd and Jacka, 1989). Basal sliding is modelled by using

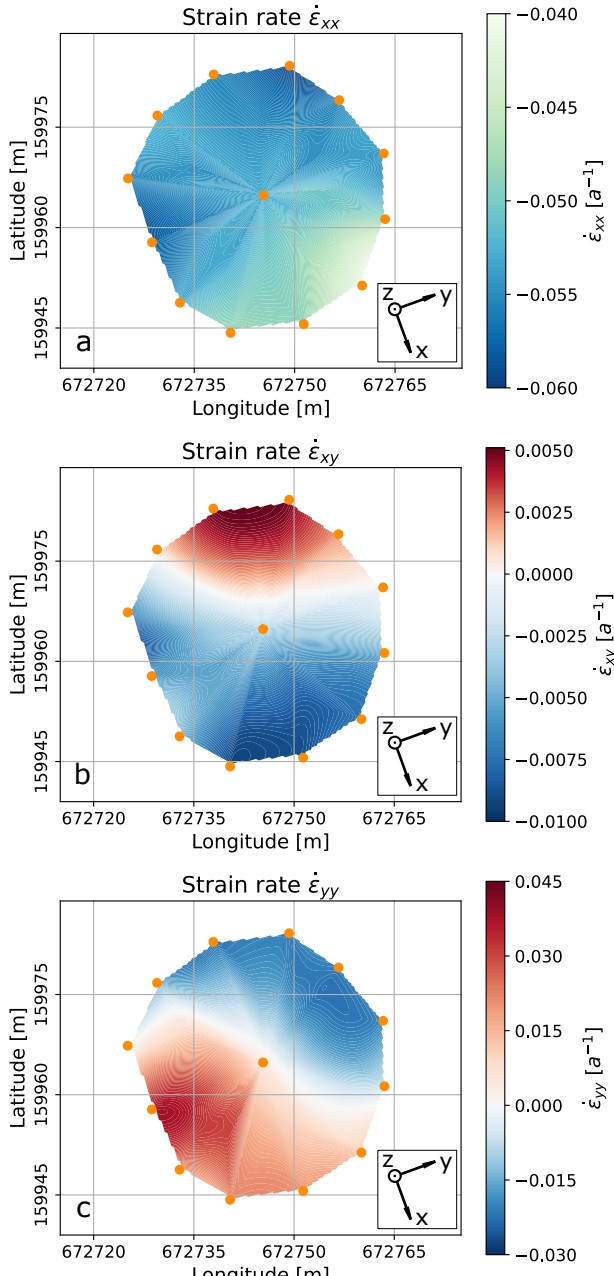

**Figure 3.** Strain rate components at the surface derived from the ice flow velocity pattern shown in Fig. 2b, (a) $\dot{\epsilon}_{xx}$, (b) $\dot{\epsilon}_{xy}$, (c) $\dot{\epsilon}_{yy}$; compressional strain rates = negative values, boreholes = orange dots, insets depict the glacier flow coordinate system.

Weertman's friction law as boundary condition at the ice-bedrock interface:

$$u_b = c\tau_b^{\frac{1}{m}} \tag{2}$$

where $u_b$ is the norm of the basal velocity, $\tau_b$ is the basal shear stress, while $m = 3$ and $c$ are constant parameters. The main model parameters are the coefficient $c$ and the rate factor $A$ that control the amount of basal motion and internal deformation, respectively. As rate factor, we choose $A = 100\,\text{MPa}^{-3}\text{a}^{-1}$, which was proved to correctly reproduce the velocities of Aletschgletscher (Jouvet et al., 2011). The rate factor is close to the literature value for temperate ice ($76\,\text{MPa}^{-3}\,\text{a}^{-1}$) (Cuffey and Paterson, 2010). As sliding coefficient, we used $c = 10\,\text{km}\,\text{MPa}^{-1}$, which was tuned such that the observed and modelled ice velocities match at the surface of the borehole. Additional sliding velocity values with less than $5\,\text{m}\,\text{a}^{-1}$, estimated from borehole camera investigations (Gräff et al., 2017), supports our assumptions for this value. It is rather small compared to previous studies on Alpine glaciers (Jouvet and Funk, 2014; Compagno et al., 2019). Here, we ran the model in a stationary fashion without time evolution. As model outputs, we obtained the velocity and stress field in three dimensions. For a sensitivity analysis, we also tested different rate factors and basal sliding coefficients. As the model still slightly overestimated the derived basal velocities, we also analysed the case without basal sliding ($c = 0\,\text{km}\,\text{MPa}^{-1}$). In this case, an extremely high rate factor $A = 200\,\text{MPa}^{-3}\text{a}^{-1}$ was required to fit the measured velocities. Furthermore, the principal stress axes did not change significantly. These changes lead to a slightly enhanced longitudinal simple shear component and slightly weaker longitudinal compressional component in the deepest parts of the ice. Therefore, we only considered $A = 100\,\text{MPa}^{-3}\,\text{a}^{-1}$ and $c = 10\,\text{km}\,\text{MPa}^{-1}$ in the following analysis.

## 4   Crystal Orientation Fabric Analysis

For detailed structural investigations of the temperate glacier ice, we performed a COF analysis in the laboratories of the Alfred-Wegener-Institute Helmholtz Centre for Polar and Marine Research (AWI). We measured the orientation of the c-axes of the ice grains. The c-axis is the symmetry axis perpendicular to the basal plane of a hexagonal crystal. Along the c-axis, the physical properties differ significantly from any direction parallel to the basal plane (the a-axes). This results in an anisotropic viscous response of the glacier ice (Schulson and Duval, 2009, Chapt. 6). Furthermore, the elastic parameters of the ice, such as bulk or shear modulus, have enhanced values in the c-axis direction and the crystal is more resistant to deformation (Cuffey and Paterson, 2010, chapter 3). This anisotropy affects the elastic properties leading to velocity changes for acoustic waves which travel through the ice (e.g. Diez and Eisen, 2015).

From the ice core extracted from the central borehole BH00 (Fig. 2a), seven samples at depths of 2, 22, 33, 45, 52, 65 and 79 m were considered. Due to technical problems during the core retrieval, the azimuthal orientations of the samples at 2 and 45 m depth are subject to some uncertainties. Their azimuthal orientations were thus obtained from extrapolations from adjacent measurements.

Each of the seven samples consisted of an ice core segment of about 50 cm length. Up to four 11 cm long adjacent sub-samples (Fig. 4) were prepared from each of these segments. Each sub-sample was then further dissected into a horizontal and two vertical cuts. All three cuts are perpendicular to each other (Fig 4). This resulted in a horizontal circular slice and two vertical slices with SN- and EW-orientations from which thin sections were prepared. We measured between 8 and 12 thin sections per sample and 77 thin sections in total. This procedure enabled a more comprehensive analysis of the large crystals existing in

temperate glacier ice (e.g. Kamb, 1959; Rigsby, 1960) and a tracing of fissures (also called fracture traces), for instance from potential meltwater intrusions. The dimensions of the pieces were $10\times6\,\mathrm{cm}$ for the vertical sections and a diameter of $\approx7\,\mathrm{cm}$ for the circular horizontal sections. The creation of sub-samples and choosing three different types of sections (horizontal, EW-vertical, SN-vertical) for every sub-sample resulted in a comprehensive analysis of at least 300 grains for each depth level. During the preparation of the ice thin sections, we also took large-area scanning macroscope (LASM) images (Binder et al., 2013; Krischke et al., 2015) from the polished surface of the $1\,\mathrm{cm}$ thick ice samples. These images provide information on the grain boundary network, subgrain boundaries, as well as the air bubble distribution, since light from the active camera is backscattered to a great extend by the evenly polished ice surfaces. Uneven parts, such as air bubbles or grain boundaries, reduce the amount of backscattered light and appear darker in the image.

All sections were analysed using cross-polarised light (Wilson et al., 2003; Peternell et al., 2009). We used the automatic fabric analyser G50 from Russell-Head Instruments (Wilson et al., 2003) to measure the orientation of the c-axis on a predefined mesh grid with a pixel resolution of $20\mathrm{x}20\,\mu\mathrm{m}^2$. The orientation of the c-axis of an ice crystal is determined by two angles:

$$\boldsymbol{c}(\vartheta,\varphi) = [\cos(\vartheta)\sin(\varphi),\ \sin(\vartheta)\sin(\varphi),\ \cos(\varphi)]. \tag{3}$$

The first angle defines the azimuth $\vartheta \in [0, 2\pi]$ of the c-axis in the horizontal plane. The second angle is the colatitude $\varphi \in [0, \frac{\pi}{2}]$ from vertical.

For the postprocessing of the obtained crystallographic data, we used the software *cAxes* (Eichler, 2013). *cAxes* analyses the misorientation angle between the determined c-axis orientations of neighbouring pixels and combines those with a misorientation $<1°$ to individual ice grains with a mean c-axis azimuth and colatitude per grain. The minimum grain size, calculated from the number of pixels multiplied by the pixel resolution, was set to $0.2\,\mathrm{mm}^2$ (500 pixels). *cAxes* automatically rotated the vertical thin sections around the horizontal x-axis $x'$ of the local measurement coordinate system $(x', y', z')$ by $90°$, so that the vertical component $z'$ is actually pointing upwards ($z' = z$, cf. Fig. 4). Then, the thin sections were rotated (with individual angles for the different types of thin sections, cf. Fig. 4) around the vertical axis $z$ to align them with the glacier flow coordinate system $(x, y, z)$. In a final step, we used the magnetometer data to correct for offsets between the theoretical and actual azimuthal orientation for the respective thin sections of the individual ice core segments. This step is necessary since some segments have slightly been rotated during retrieval, which we could reconstruct with the respective magnetometer data. This procedure ensured an identical coordinate reference frame for all thin sections along the entire ice core. Finally, we computed the eigenvalue distribution according to the procedure of Wallbrecher (1986). The three eigenvalues $\lambda_i$ (i=1, 2, 3) follow the relations $\sum\lambda_i = 1$ and $0 \leq \lambda_3 \leq \lambda_2 \leq \lambda_1 \leq 1$. These eigenvalues represent the main axes of an ellipsoid that presents the best fit for a given c-axis density distribution.

## 5 Results

Figure 5 shows the results of the COF analysis (left panels) and selected images of horizontal thin sections (right panels) for each depth level. The COF results are displayed in form of Schmidt equal-area stereo plots on the lower hemisphere (vertical

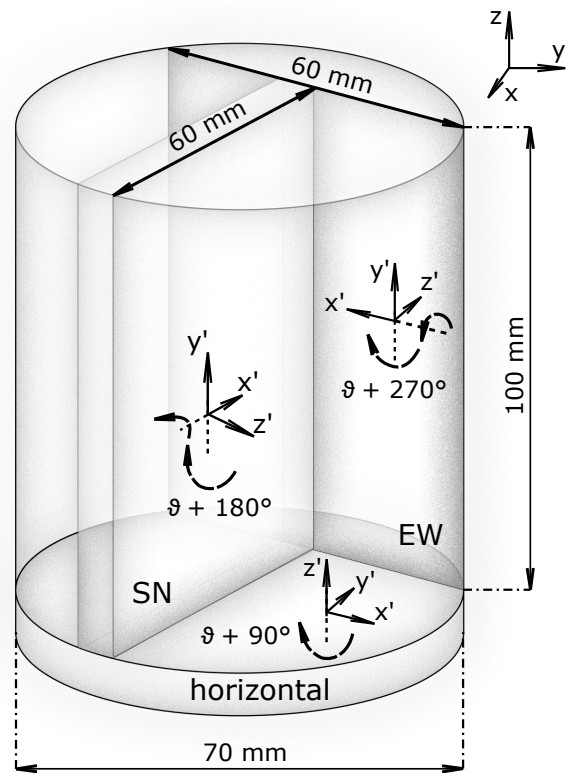

**Figure 4.** Cutting scheme for the ice core analysis. An 11 cm long piece of the ice core was cut in a horizontal (d=7 cm) and two vertical (east-west, south-north oriented, 10x6 cm) thin sections. Four thin sections of each type were analysed and combined per sampling depth.

core axis in centre). Results from all sub-samples and section orientations are combined. Each ice grain c-axis is represented by a dot. As shown by the images in the right panels of Fig. 5, the ice matrix is dominated by a few extremely large grains. Nevertheless, several hundred small grains appear along the grain boundaries or in specific patches. Especially the samples from 22 m and 45 m contain a large number of small grains. These grains form specific patterns looking like fracture traces, which are traceable through several thin sections.

For better visualising the c-axis distributions, a smoothed colour density plot, calculated in accordance to Kamb's method (Vollmer, 1995), was superimposed on the stereo plots. These density plots only consider the number of grains within the area of the stereo plot, i.e. the size of the individual grains does not affect the colour code. All density plots indicate a multi-maxima pattern and the majority exhibits four maxima. The orientation of the patterns varies with depth, but the structure inside the patterns is remarkably similar. The four maxima lie on a small circle girdle, which is characterised by an opening angle around a central vector, shown as a midpoint in the stereographic projection. Two maxima always lie on opposite sides of this midpoint and the other two on a line perpendicular to the first two clusters so that the azimuthal separation of the maxima is 90°.

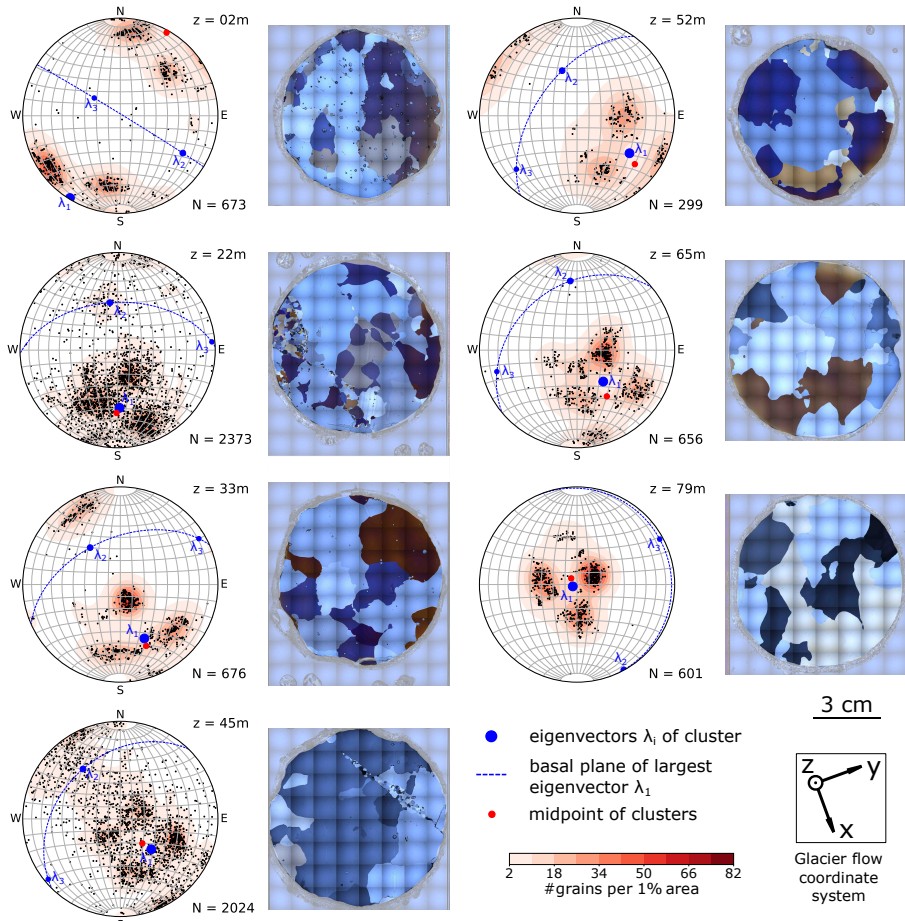

**Figure 5.** Left columns: Stereo plots (lower hemisphere Schmidt equal-area projection into the horizontal, i.e. long axis of core plots in the centre) with the c-axis distribution and associated horizontal thin sections, illustrating the typical grain size distribution, are shown for each sample. The total number of ice grains ($N$) is specified for each sample (consisting of at least 3 horizontal and 6 vertical thin sections, all rotated to horizontal view and common geographic coordinates). The sampling depth ($z$) is indicated at the upper right corner of the stereo plots. The colour code (smoothed Kamb's distribution (Vollmer, 1995)) emphasises the existing clusters of the c-axis distribution. The largest eigenvector for the determined distribution is depicted as blue dot and its normal plane is shown as dashed line. Right columns: example images of horizontal thin sections, recorded under cross-polarised light, pale edges around the actual samples are excluded from analysis.

Deviations are observed at 45 m depth, where a fifth maximum at the horizontal margin appears, and at 79 m, where one of
the four maxima is significantly weaker than the others. Depending on the number of grains per cluster the midpoint (red point
in Fig. 5) differs from the actual centroid (blue dot $\lambda_1$ in Fig. 5) of the multi-maxima pattern. The opening angle between the
midpoint and the individual maxima varies with $\pm 15°$ around a mean of $30°$ (Table 1), but the mean value is constant over all
depths.

The eigenvectors of the polycrystalline orientation tensor were calculated for each depth, and they are also shown in the stereo
plots (blue dots in Fig. 5). For enhanced visibility, the plane normal to the eigenvector associated with the largest eigenvalue
$\lambda_1$ is indicated with a dashed blue line. This eigenvector coincides with the centroid of the four-point-maxima. The other two
eigenvalues are significantly lower than $\lambda_1$ ($0.56 \leq \lambda_1 \leq 0.7$) and usually in a range of $0.1 \leq \lambda_2 \leq 0.31$ and $0.09 \leq \lambda_3 \leq 0.13$,
respectively.

With an exception for the uppermost depth at 2 m and the lowermost depth at 79 m, the azimuth of the maximum eigenvec-
tors ($147° \pm 31°$ is aligned with the direction of the glacier's ice flow ($155° \pm 10°$, cf. Figs. 1, 2). With increasing depth, the
maximum eigenvector has a decreasing colatitude, and at 79 m this eigenvector as well as the centroid of the cluster is almost
vertically oriented.

**Table 1.** Angles between individual maxima and the centroid, i.e. the largest eigenvector, describing the relative geometry of the multi-
maxima pattern and the absolute orientation (azimuth/colatutide) of the centroid.

| | relative angles within the cluster | | | | | absolute position of cluster | |
|---|---|---|---|---|---|---|---|
| depth [m] | angles per maximum [°] | | | | mean angle [°] | azimuth [°] | colatitude [°] |
| 2 | 23.3 | 27.4 | 34.3 | 34.5 | $29.9 \pm 4.7$ | 211.5 (151.5) | 88.6 |
| 22 | 23.5 | 25.3 | 34.5 | 35.2 | $29.6 \pm 5.2$ | 178 | 49.4 |
| 33 | 20.8 | 26.0 | 41.0 | 47.6 | $33.9 \pm 10.9$ | 156 | 50.6 |
| 45 | 26.9 | 29.2 | 30.7 | 33.2 | $30.0 \pm 2.3$ | 134.4 | 36.6 |
| 52 | 22.2 | 27.9 | 29.6 | 38.2 | $30.6 \pm 5.7$ | 125.6 | 55.7 |
| 65 | 20.6 | 22.5 | 34.7 | 37.6 | $28.9 \pm 7.4$ | 140.2 | 34.5 |
| 79 | 20.2 | 25.2 | 34.3 | 34.6 | $28.6 \pm 6.1$ | 246.6 | 3.9 |

For an enhanced and statistically significant data set, we combined the determined c-axis orientations, measured in up to twelve
individual thin sections with three different orientations. However, the results from the different orientations of the sections
(Fig. 4) may be inconsistent. Although the grains are not elongated in a certain direction (i.e. do not show a shape preferred
orientation), some of them appear branched and interlocked (Fig. 5, right panels). Therefore, two-dimensional cuts through
large, branched grains may let them appear as several individual grains within the same section. Kamb (1959) and Hooke
(1969) have already discussed the statistical relevance of these branched grains. The sketch in Hooke and Hudleston (1980),
Fig. 6, and more recently in Monz et al. (2020), Fig. 3, further illustrate this issue that could result in over-represented clusters
in the superimposed stereo plots from the different sub-samples.

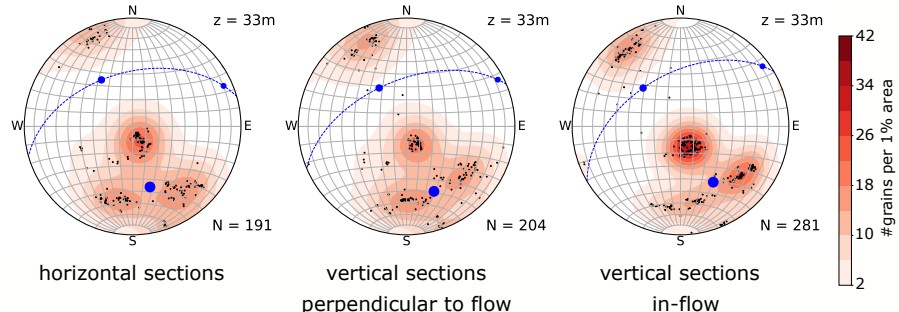

**Figure 6.** Stereo plots (lower hemisphere Schmidt equal-area projection) for the three types of sub-samples (horizontal, east-west, south-north, from left to right) with annotations as in Fig. 5.

To check the consistency of the individual orientations, the c-axis distribution for each sub-section (horizontal, east-west and south-north) was analysed separately. Figure 6 shows the results for the sample at 33 m depth. All three sub-sections show a similar pattern. The individual maxima appear in all sections and are not a result of stitching differently orientated sections together. However, due to the afore-mentioned reasons, the actual grain size is difficult to determine. Individual analyses for the other depths showed similarly consistent results (not shown).

**6   Interpretation**

    For our interpretation, we refer to the deviatoric stress tensor elements. The absolute values are shown in Fig. 7 ($\sigma_{xx}^{(d)}$ and $\sigma_{xy}^{(d)} < 0$). The x-component of the tensor elements is aligned with the longitudinal direction, i.e. the glacier flow, and the y-component is aligned with the transverse direction. Due to the flow evolution of the ice grains through the glacier, these grains are deformed under given stress conditions (Schulson and Duval, 2009, Chapt. 5). As the glacier changes its flow direction, the

ice crystals experience changing stress conditions leading to variations in the deformation regime. As a result, the c-axes of the ice grains are oriented in specific patterns, such as the multi-maxima structure that we observed in the current ice core. Stress and strain rate are directly linked via Glen's flow law and changing stress causes a change in deformation geometry. However, the particular orientation of ice grains is crucial as to whether the ice is easy to deform ("soft" direction) or whether it is further resistive ("hard" direction) against the currently applied deformation. For a detailed analysis of the current stress conditions at

the ice core location, we use the ice flow model to derive the the deviatoric stress tensor. The deviatoric stress tensor $\sigma^{(d)}$ is derived from the normal stress tensor $\sigma$ by subtracting the isotropic pressure $p$ from its main diagonal elements, i.e.

$$\sigma_{ij}^{(d)} = \sigma_{ij} - p\delta_{ij} \qquad (4)$$

where $\delta_{ij} = 1$ for $i = j$, and $\delta_{ij} = 0$ otherwise. For the deformation of the ice grains only the deviatoric stress tensor is important. The two most relevant components at the core location are the longitudinal compressional stress $\sigma_{xx}^{(d)}$ and the longitudinal shear

stress $\sigma_{xz}^{(d)}$ (Fig. 7). $\sigma_{xx}^{(d)}$ is the most dominant stress close to the surface. Its strength slightly decreases with depth. In addition,

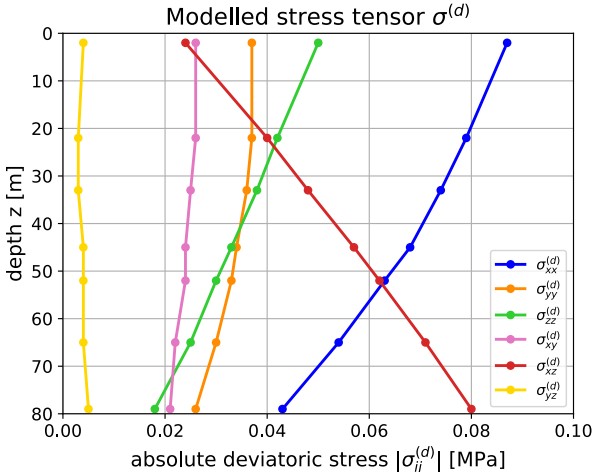

**Figure 7.** Absolute values of the deviatoric stress tensor elements obtained from ice flow modelling (x – longitudinal, y – transverse, z – vertical direction).

the shear stress $\sigma_{xz}^{(d)}$ governs the stress conditions in deeper parts as it increases with depth. The transverse components ($\sigma_{yy}^{(d)}$, $\sigma_{xy}^{(d)}$ and $\sigma_{yz}^{(d)}$) are more or less equal in all depths. We also calculate the strain rates $\dot{\epsilon}_{ij}$ from the stress tensor by using Glen's

**Table 2.** Strain rates derived from ice flow modelling (x – longitudinal, y – transverse, z – vertical direction).

| depth [m] | $\dot{\epsilon}_{xx}$ [a$^{-1}$] | $\dot{\epsilon}_{yy}$ [a$^{-1}$] | $\dot{\epsilon}_{zz}$ [a$^{-1}$] | $\dot{\epsilon}_{xy}$ [a$^{-1}$] | $\dot{\epsilon}_{xz}$ [a$^{-1}$] | $\dot{\epsilon}_{yz}$ [a$^{-1}$] |
|---|---|---|---|---|---|---|
| 2 | -0.061 | 0.026 | 0.035 | -0.018 | 0.017 | 0.002 |
| 22 | -0.056 | 0.026 | 0.030 | -0.018 | 0.028 | 0.002 |
| 33 | -0.053 | 0.026 | 0.027 | -0.018 | 0.034 | 0.002 |
| 45 | -0.049 | 0.025 | 0.024 | -0.018 | 0.041 | 0.003 |
| 52 | -0.047 | 0.024 | 0.023 | -0.017 | 0.046 | 0.003 |
| 65 | -0.042 | 0.023 | 0.019 | -0.017 | 0.055 | 0.003 |
| 79 | -0.036 | 0.021 | 0.014 | -0.017 | 0.065 | 0.004 |

flow law, i.e. Eq. (1) (Table 2). Since the model does not consider anisotropy, orientation dependent response of the anisotropic ice is not considered. For typical fabrics (e.g. single maximum, girdle fabric), enhancement factors could be introduced (e.g. Thorsteinsson, 2001; Pettit et al., 2007; Ma et al., 2010) to overcome this issue. Beyond, anisotropic flow laws (e.g. Gillet-Chaulet et al., 2005) can be employed to consider more complex fabrics such as the multi-maxima pattern. Furthermore, in isotropic models, stress and strain rate are connected with a scalar value and the principal axes of stress and strain rate tensors are parallel. These are a crucial limitation to be considered in the following interpretation. Especially for shear stress, the model-derived and the actual strain rate directions may differ significantly. In addition, the quantitative numbers deviate from


isotropic ice as the aforementioned basal sliding affects the strain rates that an ice grain experiences. However, we regard the modelling output as auxiliary values for our interpretation.

In the following, we provide an interpretation of three significant features presented in Fig. 5, namely

- the azimuthal orientation of the c-axes distributions as represented by the maximum eigenvectors of the stress tensors,

- the decrease of the maximum eigenvector colatitudes with increasing depths (viz. this eigenvector becomes more verti-
cal), and

- the existence of multi-maxima patterns in the c-axis distributions.

To support the interpretation, the stereo plots in Fig. 5 are shown again in Fig. 8 with adjustments as a result of the following interpretation. Here, only the colour-coded c-axis density distributions are plotted, superimposed by additional information obtained from accompanying analyses.

## 6.1  Azimuthal orientation

Results from all depth levels, with the exception for $2\,\mathrm{m}$ (Fig. 5), show in general a mean c-axis orientation (Table 1) approximately parallel to the main glacier flow direction that was obtained from the displacements of the surrounding boreholes (Fig. 2, $\approx 155°$). The largest eigenvector $\lambda_1$ always lies in a vertical plane that is aligned ($\pm\,20°$) with the glacier flow direction. This flow kinematics, depicted by the principal stress axis (Figs. 5 and 8) is associated with the alignment of the centroid of the
c-axes. As this flow direction changes, the COF have most likely developed since the glacier flows in the direction observed at our drill location. Although the centroid in $79\,\mathrm{m}$ is vertically oriented, the characteristic "diamond shape" of the multi-maxima pattern is still joining the vertical plane of the glacier flow direction (the verticality of the pattern is discussed below).

The observed azimuthal alignment of the COF with the glacier flow (with some limitations for $79\,\mathrm{m}$) is in accordance with results from laboratory experiments in a number of previous studies (e.g Kamb, 1972; Duval, 1981; Budd and Jacka, 1989)
and comparable with some parts of the Cape Folger ice core (Thwaites et al., 1984).

The uppermost sample ($2\,\mathrm{m}$) does not fit into this interpretation. Although the magnetometer data are consistent, the core break between two segments at $10\,\mathrm{m}$ was unclear and we cannot fully exclude a misorientation between the segments in 2 and $22\,\mathrm{m}$. As shown in Fig. 8 an azimuthal correction of $-60°$ would lead to a perfect alignment of this sample with all other observations and the modelling results for the particular location. Therefore, we assume a misorientation of the core segments.

## 6.2  Colatitude variations

In the following, we consider the variations of the colatitude of the largest eigenvector $\lambda_1$ (Fig. 8, blue dot). There is a decrease of the colatitude from $89°$ to $4°$ with increasing depth (Table 1). Considering the deformation mechanisms, mainly dislocation creep, this is the result of the stress and strain rate distribution (Fig. 7 and Table 2) in the glacier. The ice crystal c-axes in our samples generally orient themselves parallel to the ice flow, which coincides with the modelled maximum compressional
principal stress direction ($\sigma_1$ in Fig. 8). As indicated by the relative movements of the surrounding boreholes (Figs. 2 and 3), we

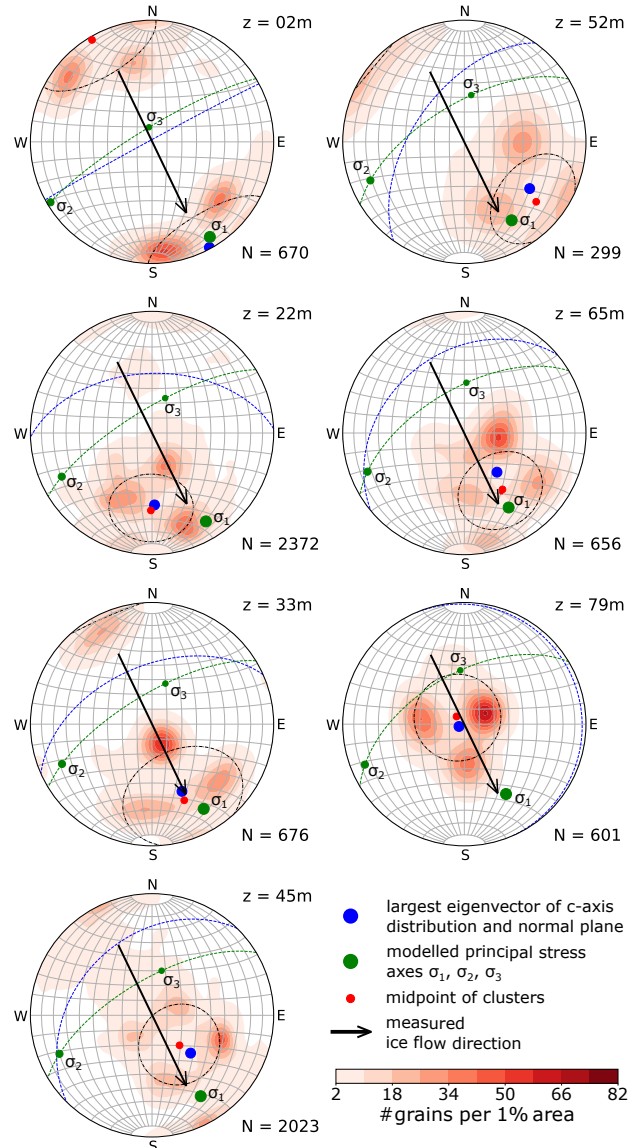

**Figure 8.** Stereo plots (lower hemisphere Schmidt equal-area projection) with the colour coded (same as Fig. 5) c-axis distribution for each sample are shown. Note: The azimuth for z=2 m is corrected by -60° (see text for discussion). The total number of ice grains is specified for each sample. The black dashed line shows the mean opening angle for the cone of maxima around the centroid depicted as red dot. The calculated largest eigenvector for the c-axis distribution is shown as blue dot (its normal plane as dashed blue line) and the calculated largest compressional principal stress axis from the ice flow model is represented by a green dot (its normal plane as dashed green line).

indeed observe a compression at the glacier surface. In accordance with borehole inclination measurements, the flow-parallel compression also occurs at greater depths but slightly decreases. In contrast, with increasing depth, the shear stress significantly increases. Especially in the deepest parts of the ice core, the longitudinal shear component $\sigma_{xz}^{(d)}$ governs the stress state, which is also confirmed by the inclinometer measurements. This increasing shear deformation of the ice lets the colatitude angle decrease with increasing depths, and explains, at least qualitatively, the observations in Fig. 8.

The principal stresses ($\sigma_1, \sigma_2, \sigma_3$), derived from the ice flow model, were added to Fig. 8 (green dots) for comparison. Although not matching perfectly, the colatitudinal angles of the largest eigenvector $\lambda_1$ and the dominant principal stress $\sigma_1$ are similar within $\pm 26°$ with an exception for the deepest sample. The discrepancy especially for the deepest sample is considered as evidence that the c-axis distribution is governed by strain and not stress in the last consequence (Budd et al., 2013; Faria et al., 2014b; Weikusat et al., 2017). In the presence of simple shear, the direction of the principal stress axis and principal strain rate axis are not aligned (non-coaxial relation) Cuffey and Paterson (2010, Chapt. 3). This implies that the COF for the deepest sample is dominated by the shear component, which approaches simple shear. According to our strain rate components (Table 2) such an implication is justified. The modelled shear strain rate is twice as large as any other component and causes the most significant effect in the borehole measurements after a short time period.

### 6.3 Multi-maxima c-axes distribution

If the c-axis orientations would be governed solely by the orientation of the major principal stress and strain rate direction ($\sigma_1$) (mainly a result of compressional and simple shear stress), we would rather expect a single maximum in the stereo plots as in deeper parts of other ice cores (Faria et al., 2014a). As observed in Figs. 5 and 8, there is no single maximum. Instead, the individual c-axes in our samples deviate on average about $30°$ from the principal stress or strain rate (for $79\,\mathrm{m}$) direction (indicated by black small circle girdles in Fig. 8) and group in several maxima.

The most likely reason for this observation involves recrystallisation processes. In general, two types of recrystallisation have been considered, namely rotation recrystallisation (RRX) and migration recrystallisation (Alley et al., 1995). RRX (described in detail in Alley (1988)) counteracts against the dynamic grain growth (Weikusat et al., 2011b). According to Faria et al. (2014b), migration recrystallisation, also called strain-induced boundary migration (SIBM), needs to be subdivided into two types, namely SIBM-O and SIBM-N. In both cases, grains with less dislocations grow at the cost of grains with a large amount of dislocations. The first type assumes that under given strain rate conditions, already existing (i.e. old), suitably oriented grains grow at the cost of less suitably oriented grains (called SIBM-O). The second type is very similar, but with a relevant difference: when the grain grows parts at the boundaries like bulges can be nuclei for new smaller grains (therefore called SIBM-N, see an exemplary process described by Steinbach et al. (2017) with a very similar orientation like their parent grain). These new grains are considered as strain-free grains and have an impact on the grain size. Both mechanisms are driven by reducing the thermodynamic energy of the whole system and grains with heterogeneously distributed dislocations are absorbed (Weikusat et al., 2009b, Fig. 8).

In our data set we observe a variety of different grain sizes. Table 3 summarises the grain size distribution in our ice core samples. As mentioned earlier, the two-dimensional cuts through the ice core samples may lead to a misinterpretation of the

**Table 3.** Grain size distribution (median, mean, and maximum grain sizes [mm$^2$], minimum grain size is 0.2 mm$^2$ and defined as threshold during processing) and number of grains within defined grain size classes [mm$^2$].

| depth [m] | number of grains | median | mean | max | number of grains per class | | | | | |
|---|---|---|---|---|---|---|---|---|---|---|
| | | | | | <1 | 1-5 | 5-20 | 20-100 | 100-500 | >500 |
| 2 | 673 | 5.30 | 77.06 | 1826.71 | 211 | 123 | 102 | 135 | 76 | 26 |
| 22 | 2373 | 0.79 | 19.04 | 1551.97 | 1340 | 584 | 192 | 165 | 69 | 23 |
| 33 | 676 | 4.41 | 85.97 | 3994.17 | 202 | 146 | 110 | 106 | 83 | 29 |
| 45 | 2024 | 0.88 | 29.89 | 3249.29 | 1078 | 522 | 173 | 133 | 91 | 27 |
| 52 | 299 | 14.67 | 121.42 | 1986.70 | 52 | 56 | 49 | 66 | 55 | 21 |
| 65 | 656 | 6.01 | 95.02 | 3752.65 | 195 | 119 | 123 | 112 | 67 | 40 |
| 79 | 601 | 6.81 | 96.79 | 3236.18 | 141 | 134 | 109 | 111 | 79 | 27 |

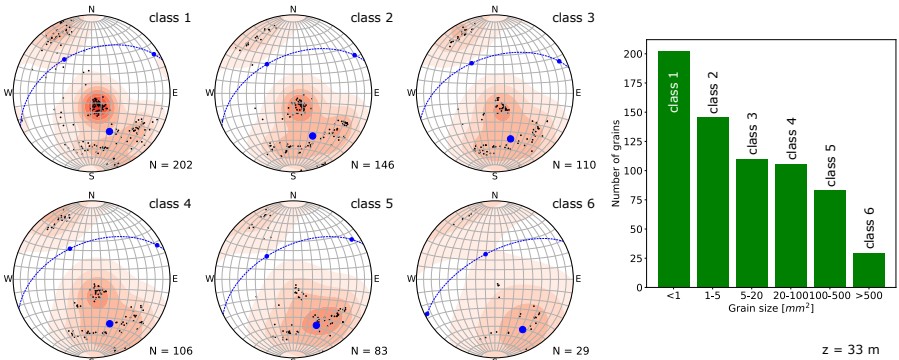

**Figure 9.** Stereo plots (lower hemisphere Schmidt equal-area projection) with the colour coded (same as Fig. 5). Each plot represents a grain size class as indcated at the upper right corner. The number of grains per class is specified for each plot. The calculated largest eigenvector for the c-axis distribution is shown as blue dot (its normal plane as dashed blue line). Right column: histogram showing the number of grains per class.

grain sizes as large interlocked grains can appear as several small grains. However, it can be expected that not all small grains are cut branches of large grains. We also analysed the COF for six different grain size classes individually. The data are shown in Fig. 9 for one sample (other samples in the supplement). The multi-maxima pattern is persistent in all grain size classes. Based on these findings, we postulate that this pattern is a result of SIBM-N in combination with the described longitudinal compressional and shear stress.

However, our data set cannot conclusively explain the very regular distribution of c-axes, i.e. the "diamond shape", within the multi-maxima pattern.

## 7 Discussion

The literature on COF field studies is not conclusive concerning the existence of multi-maxima fabrics. They were observed in early studies on temperate glaciers (e.g., Rigsby, 1951; Kamb, 1959; Rigsby, 1960) for the first time. In the 1970's to 1980's, they have been found in ice caps with ice temperatures above -10°C (Hooke and Hudleston, 1980), and also in the bottom ice of Byrd Station and Cape Folger in Antarctica (Gow and Williamson, 1976; Thwaites et al., 1984). At that time, the estimation of c-axis distributions was more subjective and could not benefit from modern equipment. The orientation of crystals was determined manually on a Rigsby-stage by turning and tilting the ice samples between polarised plates. Thus, only a limited number of grains (up to 100) and usually the largest grains were analysed. Therefore, the "diamond shape" pattern was often debated to be a statistical effect.

Interestingly, in more recent studies on other temperate glaciers multi-maxima fabrics in combination with a large grain size were only observed in the deepest parts of the ice cores drilled in the ablation zone (e.g. Tison and Hubbard, 2000). The conditions for a "diamond shape" pattern seem to be suitable in large glaciers like the Rhonegletscher with its high temperatures and large ice flow velocities compared to other valley glaciers. In polar ice cores, a multi-maxima fabrics has been observed only in the deepest parts of some Antarctic and Greenlandic cores (Gow and Williamson, 1976; Thwaites et al., 1984; Montagnat et al., 2014). There, the temperature conditions are as high as in temperate glaciers like the Rhonegletscher.

Laboratory experiments, performed on artificial ice under high temperatures (> -2°C), provide evidence for two features we observe. Maohuan et al. (1985) created multi-maxima fabrics with combined shear and compressional stresses in their torsion-compression-experiments. The pattern clustered around the maximum principal stress direction. In addition, Jacka and Maccagnan (1984) analysed the opening angle of small circle girdles formed under compressional stress. The ranges for the opening angles are identical with our observations. For such opening angles between compressional direction and the c-axis direction, the compressive strength applied onto the ice crystal is minimised (Schulson and Duval, 2009, Chapt. 11).

As a result of these laboratory experiments and field measurements, we conclude that these temperate conditions in Rhonegletscher are a prerequisite for the development of multi-maxima patterns, but a multi-maxima pattern is not necessarily found in all temperate glaciers.

Recrystallisation has regularily been observed in a variety of ice cores (Alley, 1988; Duval and Castelnau, 1995; Weikusat et al., 2009b; Schulson and Duval, 2009; Cuffey and Paterson, 2010; Faria et al., 2014b). Recrystallisaion has usually been divided into two types – rotation and migration recrystallisation. However, according to Faria et al. (2014b), migration recrystallisation, also called strain-induced boundary migration (SIBM), needs to be subdivided into two types (SIBM-O and SIBM-N). These two mechanisms lead to different COF. If SIBM-O is the dominating process, a single maximum consisting of a few very large grains is expected to develop. For SIBM-N we regularly generate new grains with a similar orientation like the parent grain and concurrent maxima are likely to develop. This can lead to the observed multi-maxima fabric. Evidence for a dominating grain boundary migration with nucleation (SIBM-N) can be observed in LASM scans (Fig. 10). Firstly, the air bubbles are trapped within grains with a diameter of centimetres. Secondly, the grain boundaries of neighbouring grains bulge and smaller grains develop either as "island grains" or as clusters along the boundaries of the large grains (blue, red and yellow circles in Fig. 10).

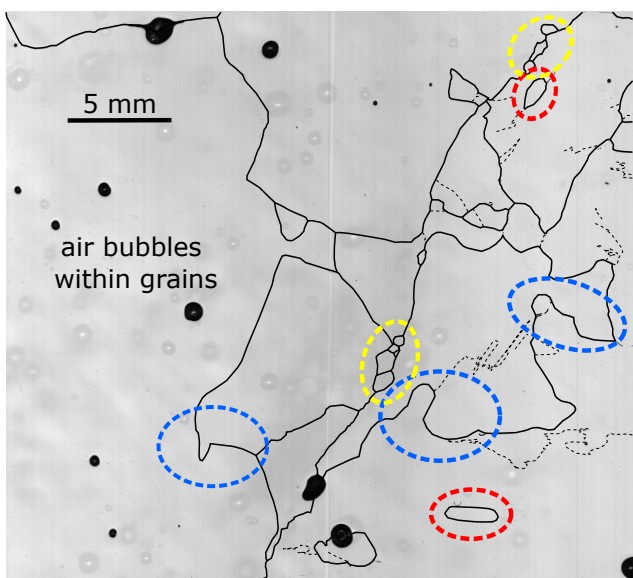

**Figure 10.** Example of a LASM image indicating several processes in the temperate ice: black areas are air bubbles, black lines represent the grain boundaries, and black dashed lines indicate subgrain boundaries. Air bubbles can be found completely trapped within individual large ice grains indicating migration recrystallisation. Bulging of grain boundaries (blue circles), the development of island grains (red circles) and small grains along the boundaries of large grains (yellow circle) provide evidence for a dominating SIBM-N-type recrystallisation.

The trapped air bubbles and newly developing grains provide evidence that SIBM-N is the dominating process in these samples of temperate ice.

The recrystallisation does not conclusively provide arguments for the regular "diamond shape" within the multi-maxima pattern. One potential mechanism to be considered are localisation effects. In this theory, the polycrystalline system distributes

the impact of the different strain along localisation bands, especially in presence of additional aggregates such as air bubbles (Steinbach et al., 2016). This may lead to a highly variable rate of fabric change. A detailed investigation of localisation bands is rather difficult in presence of grain boundary migration (i.e. SIBM-O and SIBM-N) as recrystallisation restores the crystal shape and removes the typical shear bands (Llorens et al., 2016). Llorens et al. (2017) investigated the COF for simple shear experiments by considering localisation bands and dynamic recrystallisation (rotational recrystallisation and grain boundary

migration, but without nucleation). These strain-induced localisation effects might be worth to be considered for explaining multi-maxima patterns.

Further alternative explanations can and should be considered: Kamb (1959) calculated the preferred maximum positions of a "diamond shape" pattern with considerations of single crystal compliance constants under recrystallisation. This explanation could add missing details to our interpretation and describe the regularity in the "diamond shape" pattern. Matsuda and Waka-

360 hama (1978) suggest twinning effects that may occur when c-axes develop under recrystallisation. Potentially, these effects may lead to a clustering of the c-axes and would even better explain the regular shape of the COF. This theory is supported by

the fact that the opening angles of two opposing clusters are generally similar, whereas the angles compared to the other two maxima can vary within 15-20° and therefore called "diamond shape" fabrics. Apart from that, it is hard to find any studies about observations on twinning as result of ice deformation in glaciers and Faria et al. (2014b) summarised that mechanical twinning has not been observed in glacier ice. To investigate this further, we would need to measure the orientation of the ice crystals a-axes (in addition to the c-axes), for an enhanced image about the crystal orientation in three dimensions (Weikusat et al., 2011a; Journaux et al., 2019; Monz et al., 2020).

## 8    Conclusions

COF analyses of an ice core, extracted from a temperate alpine glacier, showed conspicuous multi-maxima patterns of the c-axes. This was observed at different depth levels. The azimuth and colatitude of the centroid of these multi-maxima patterns are well-aligned with the main principal stress direction. Close to the surface, compressional longitudinal stress conditions lead to a horizontal orientation aligned with the glacier flow. In deeper parts, the dominating longitudinal shear stress causes a vertical COF. The mean basal plane is aligned with the shear plane.

Strain-induced boundary migration with nucleation of new grains (SIBM-N) seems to be the dominating recrystallisation process under the given high temperatures and strain rates. This provides an explanation for the large and branched grains accompanied by smaller grains with similar orientation and thus a clustering in several maxima. However, the observations cannot conclusively provide an answer for the very regular "diamond shape" pattern.

To the best of our knowledge, this is the first comprehensive COF analysis of an ice core from a temperate alpine glacier that links the COF with the glacier flow. The results are consistent with supporting measurements and modelling results. These consistencies are encouraging, and will hopefully motivate similar studies on other temperate glaciers.

*Data availability.* The ice fabric data and the LASM images are published in the open-access database PANGAEA® (Hellmann et al., 2018a,b).

**Appendix A**

*Author contributions.* This study was initiated and supervised by HM, AB, IW and MS. The field and laboratory data were collected by MS, SH, MG, AB and JK and analysed by SH with support from JK and under supervision of IW and MS. Data processing and calculations were made and interpreted by SH and discussed with JK, IW, and MG. The ice flow was modelled by GJ with input data from MG. The paper was written by SH with comments and suggestions for improvements from all co-authors.

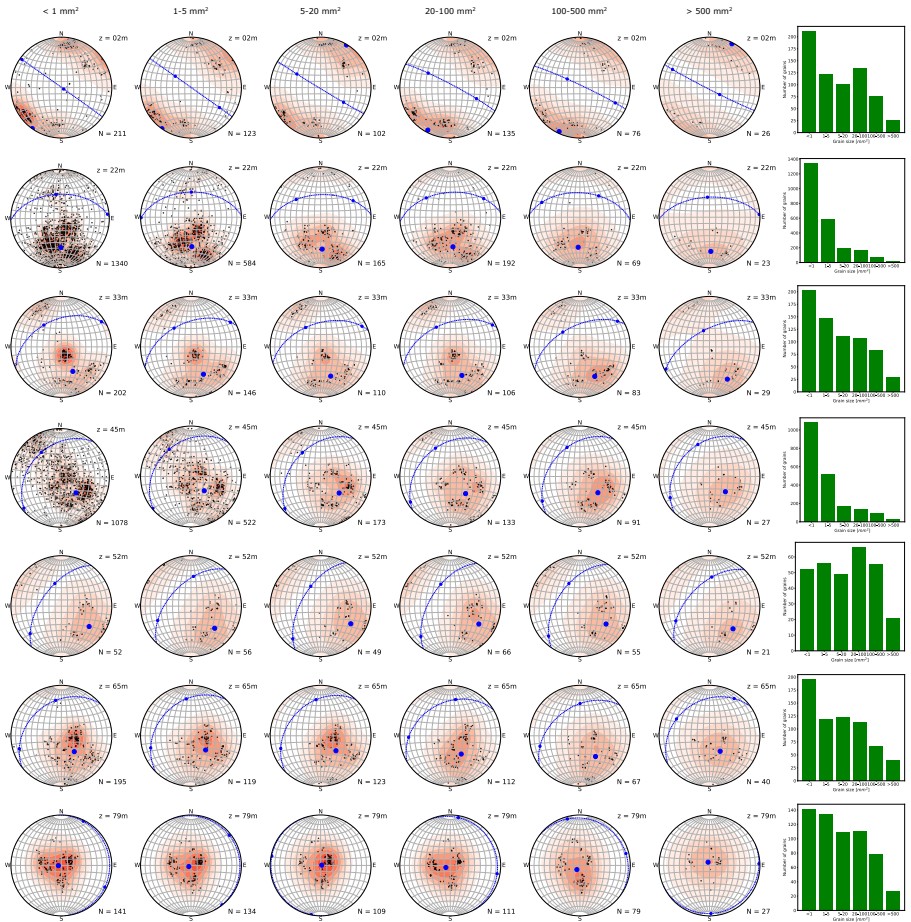

**Figure A1.** Stereo plots (lower hemisphere Schmidt equal-area projection) with the colour coded (same as Fig. 5). Each column represents a grain size class as indcated at the top. Each line represents one of the 7 samples of Fig. 5. The number of grains per class is specified for each plot. The calculated largest eigenvector for the c-axis distribution is shown as blue dot (its normal plane as dashed blue line). Last column: histograms showing the number of grains per class for each sample.

*Competing interests.* The authors declare that they have no conflict of interest.

*Acknowledgements.* This project is funded by the Swiss National Science Foundation under the SNF Grants 200021_169329/1 and 200021_169329/2.
Data acquisition has been provided by the Paul-Scherrer Institute, Villingen, the Alfred Wegener Institute Helmholtz Centre for Polar and Marine Research (AWI), Bremerhaven and the Laboratory of Hydraulics, Hydrology and Glaciology (VAW) of ETH Zurich. We especially thank J. Eichler, T. Gerber, T. Jenk, and D. Stampfli for their extensive technical and logistical support during ice core drilling and processing. Valuable discussions with the Structural Glaciology group at University of Tuebingen helped to improve the paper. We would like to thank the editor, Olivier Gagliardini, and the two reviewers, Erin Pettit and Peter Hudleston, for their constructive comments, which greatly
improved the quality of the manuscript.

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
