# Peer review of "Crystallographic analysis of temperate ice on Rhonegletscher, Swiss Alps"

_The Cryosphere, 2020_

## Referee Comment (RC1) · Peter Hudleston (Referee) · 30 Jun 2020

Review of "Crystallographic analysis ice on Rhonegletscher, Swiss Alps" by Hellman et al.

General

This paper provides a detailed description and analysis of the crystallographic fabric of ice taken from a core from the surface to bedrock in the central part of the ablation zone of a temperate valley glacier. It finds that multimaxima fabrics of the type commonly found in most valley glaciers, usually just from near-surface samples, occur at all depths within the glacier, with some systematic changes with depth in orientation of the clusters that constitute the fabric. This is a new finding and deserves to be published on this basis alone. The paper then, importantly, relates the fabric to the stress field derived from numerical modeling and finds a direct relationship between the orientation of the fabric and orientation of the modeled principal stresses. This leads to a possible explanation of these four maxima fabrics. I question parts of the interpretation and don't believe these fabrics are yet fully explained, as discussed in the specific comments below, keyed to lines in the text. I have also corrected a few typographical errors and made some suggestions for language usage.

Specific points keyed to line numbers in the text

Line 9-10. The language here doesn't clearly describe the observed relations, since there are four azimuths and colatitudes that define the fabric and three principal stress directions. It is the centroid of the fabric and the maximum principal stress direction that nearly coincide in orientation.

Line 31. The stress and kinematic conditions in valley glaciers are more complex than just combinations of simple and pure shear.

Line 94. Although the details of the numerical model need not be given here, the basic form of the flow law should be given, since the value of the flow law parameter A is defined. The value of the power law exponent, n, in the flow law should also be noted.

Line 117. It is not clear what is meant by fractures here, since there are no actual fractures in this core. This needs clarification. What are the physical manifestation of the 'fractures?" They must be defined by some combination and bubble or grain size distribution.

Line 135. Surely this is mm2 not $\mu$m2

Line 151. Here is some information about the fractures. Presumably these patterns are in the form of linear traces in thin section. Following Hambrey I like the term 'fracture traces' for these likely healed fractures.

Line 157. You use the term centroid here for the maximum eigenvector on these plots,

and state that these are equivalent in the caption to Table 1. Yet in Figure 7 the two are represented and plotted as separate entities. The usage needs to be consistent. In this case how is centroid defined?

Line 173-174. It should be noted that Kamb, Hooke and others have discussed the issue of accounting for complex and branching shapes of large grains when making c-axis plots.

Fig. 6. The caption could be shortened by stating that the annotation is as in Fig. 5

Line 192. The c-axis fabric has orthorhombic (and perhaps close to axial) symmetry, but is this also true of the stresses? What about the other two principal stresses. Are they consistent with plane strain or plane stress, as appears to be assumed in Fig. 9? Are the principal stresses and strain rates in this section of the glacier near the surface parallel to the flow direction ($\sigma 1$), vertical ($\sigma 3$) and horizontal ($\sigma 2$), with the lateral strain rate close to zero, as one would expect for a valley of constant width. One would expect the maximum principal stress to become inclined deeper into the ice as shear stress parallel to the base increases, which the modeled stress shows a tendency to do.

Line 200-204. This is a possible explanation, but I prefer the misorientation of the sample as the explanation, which as you state, fits very nicely when a 60o azimuthal 'correction' is made. A preserved fabric from earlier in the flow path is less likely at high temperatures when rapid recrystallization is expected.

Lines 210-213. With this explanation you would expect $\sigma 1$ to be vertical to explain the fabric at 79m depth and not as given by the numerical model. Although the vertical normal stress increases with depth, it is the deviatoric stress that controls deformation, not absolute stress values, and this likely does not change greatly with depth. I think the main thing that changes with depth is not the vertical effective compression ($\sigma 1 - \sigma$mean) but the increasing addition of base parallel shear stress, that in general terms increases linearly with depth.

Line 217. There is almost certainly some dependence of fabric on strain, which may not be great with fast recrystallization.

Line 213-214. In simple shear the directions of principal stress are only aligned with those of principal strain for infinitesimal strains. The divergence grows as strain increases.

Line 238 I don't believe Cuffey and Paterson really explain why there should be four maxima when the stress deviates from unconfined compression. This is more of an observation than an explanation.

Lines 244-245. This is unclear. A change in direction of glacier flow could be associated with either an increase or decrease in strain rate and thus decrease or increase in recrystallized grain size. Why just a decrease?

Line 251 and Table 2. Table 2 does not really give the grain size distribution, only average numbers of grains and average and maximum size in each sample. It would be useful to know the number of grains in each size category. Also interesting to know if there is any difference between the large and small grains in COF.

Line 255-266. I'm not sure how much information is given by the air bubbles, except they do provide excellent evidence of active recrystallization by grain boundary migration. Bubbles are found both within grains and along grain boundaries both in temperate ice and in cold ice experiencing dynamic recrystallization, although the recrystallization mechanism may differ.

Line 269. Hooke and Hudleston were concerned with polar, not temperate ice. The study was made on the Barnes Ice Cap.

Line 276. Whether the multimaxima fabrics are a result of unrepresentative sampling is still arguable in some circumstances, although the case you have here for these being true multimaxima fabrics is a strong one.

Line 290-291. I think more data is needed to support this conclusion. The cores taken

by Tlson and Hubbard were in a different regime within the glacier – accumulation zone where perhaps there is longitudinal extension rather than compression, and close to the lateral margin of the glacier rather than in the center. This must lead to a more complex stress regime.

Line 298-299. The combination of compression plus simple shear as applied in these experiments makes sense for much of your core, but not for the highest one where the shear component is minimal, nor for the lowest one, where the $\sigma 1$ direction lies well outside the small-circle girdle of maxima. Some other explanation must hold in these places.

Line 300-301. I'm not sure if I'm properly interpreting what you are saying here, but the planes of maximum shearing stress in Duval's combined compression-simple shear experiments are not vertical and horizontal in his experiments, but inclined by an amount that depends on the relative amounts of normal compression and simple shear.

Line 312-316. Both Llorens et al. and Qi et a. are dealing only with simple shear, not with combined compression plus simple shear as in the torsion plus compression experiments. The conditions in the Rhone glacier I imagine change from horizontal compression with minimal base-parallel shear near the surface to horizontal compression combined with increasing base-parallel shear near the base of the glacier. As theory shows, shear stress increases approximately linearly with depth, while longitudinal stress stays approximately constant.

Fig. 9. The stress state shown in Fig. 9 is almost that of simple shear (no base-parallel longitudinal compression) with the shear plane (taken as the glacier bed) horizontal and $\sigma 1$ inclined at 45o to the shear plane. If it is simple shear, there will be no horizontal compression and thus no shortening in the glacier flow direction, which is incompatible with your data. If horizontal glacier flow-parallel compression is added $\sigma 1$ will move closer to horizontal than it would be for simple shear alone. This looks like being the case for much of the glacier from the stresses shown in Fig. 7. I would expect the

inclination of $\sigma 1$ to be near zero at the surface and something less than 45o close to bedrock, the amount depending on the amount of horizontal compression. Although not a smooth change, the $\sigma 1$ directions in Fig. 7 are consistent with this.

The plot in Fig. 9 does not correspond to any of the plots in Fig. 7, all of which have $\sigma 1$ at a shallower inclination than 45o and thus have associated planes of maximum shearing stress that are neither vertical or horizontal, unlike the situation in Fig. 9. The one closest to horizontal thus cannot be considered a plane of simple shear. The 'shear plane' must always be the presumably sub-horizontal glacier bed.

Line 340. I disagree with the statement here (see comments for lines 210-213). Although the absolute value of the vertical normal stress increases with depth, the deviatoric vertical normal stress changes much less. It is the increase in base-parallel shear stress combined with the horizontal compressive stress ($\sigma$xx if you like) that causes $\sigma 1$ to rotate from near horizontal at the surface to inclined at some angle of less than 45o at the base.

Line 342. The second part (ii) of the explanation for multimaxima fabrics given here makes no sense by itself. All states of stress that are non isotropic involve shear stresses. If the multimaxima fabric depended solely on the state of stress – that is with instantaneous adjustment of the c-axis fabric as the stress field changes – then there should be a constant relationship between the positions of the maxima and the principal stress directions. This clearly is not the case as the relationship in the deepest sample shows. There is, however, as you note, a consistent relationship between the fabric and the $\sigma 1$ direction through most of the glacier and in all cases, with small deviations, the centroid of the COF fabric and the $\sigma 1$ direction lie in the vertical plane that contains the flow direction. This is a key relationship that I believe you have only partly explained.

Peter Hudleston

Please also note the supplement to this comment:
https://tc.copernicus.org/preprints/tc-2020-133/tc-2020-133-RC1-supplement.pdf

---

## Referee Comment (RC2) · Peter Hudleston (Referee) · 1 Jul 2020

Just to note that my comment on lines 210-213 contains an error: $(\sigma 1 - \sigma\text{mean})$ should be $(\sigma\text{v} - \sigma\text{mean})$
* * *

---

## Referee Comment (RC3) · Erin Pettit (Referee) · 27 Aug 2020

General comments This paper provides a careful analysis of the measured fabric in the central part of a Rhonegletscher, in the ablation area through ice cores (not quite to bedrock).

The authors did a really thorough and rigorous analysis of the fabric using multiple thin sections in 3 orientations. The most well analyzed core I have seen for a temperate glacier, I appreciated the thoroughness, as it was necessary because of the dominance by large grains and a grain size distribution that if far from normal. This paper in some form should be published because of the beautiful data set.

[Figure]

Interactive
comment

While the data analysis is done really well, the interpretation in terms of stress state is not as thorough and rigorous. Their qualitative interpretation of the stress state and its relation to fabric and recrystallization processes is confusing and in a few places incorrect. The paper would benefit from a summary of the key states of stress, key metamorphic processes, citing the original research (going beyond Cuffey and Paterson and the Faria reviews). As a reader, if I am to trust their interpretation of the fabric, I need to trust that they understand the underlying physics. At this point, the physics is description is still lacking. It is imperative that the interpretation of the fabric in terms of the stresses be written with the same care and rigor that that fabric was measured.

First, it would be helpful to clarify when deviatoric stress is being used versus total stress. Deviatoric stress control most of the deformation and pressure plays a minor, if any, role in deformation, therefore describe the deviatoric stress states rather than "absolute" stress and "overburden." For example, the authors suggest that there is less deformation in the surface layers because the "absolute" stress there is low, but this is not the case - the vertical compressive deviatoric stress is no necessarily smaller at the surface, it is typically about the same - it is only the pressure term in the total stress that is smaller at the surface and pressure does not drive fabric (only gradients in pressure or overburden can drive flow).

I would suggest that they re-write the description of the stress state in terms of more formal tensor components, and more specific (and correct) wording. And please be explicit about what is behavior linked to stress and what is behavior linked to strain rate (and discuss with respect to the statement that strain rate ultimately drives fabric development not stress).

In terms of writing, there are numerous run-on sentence, imprecise wording, and extensive use of passive voice, all of which slows down the reader. I provide examples of a few of these (but not all of them) below, I encourage the authors to edit carefully for these three writing issues.

[Figure]

Specific suggestions:

Line 1 Abstract - the first line of the abstract should offer some kind of bigger picture motivation, something to entice readers beyond those already rheology and fabric "geeks" - this is a neat paper with respect to the unique measurements and it would nice for the broader glaciology community to read it.

Line 20: delete "to that" - not necessary

Line 23: delete "do"

Line 24: Faria offered great reviews in his 2014 papers, but be careful citing those papers when there are better papers that are more directly or more originally related to the statement. Here by citing Faria, it implies that that paper was the first to discover that COF evolves in a glacier. Provide more direct/original citations please (or be explicit that you are citing Faria as a review article).

Line 30: delete "quickly" unless you want to provide the timescale that quickly is indicative of (words like quickly, clearly, mostly, etc don't add any information and can lead to confusion).

Line 53 - run-on sentence, breakup into two or three.

Lines 74-78 - lots of passive voice here, rewrite

Line 78 - I'm still a little confused how you knew the orientation, did the drill head not spin on the cable as it was lowered or raised?

Figure 2/3: I like the diagram in figure 3, but why not just calculate the bulk surface strain rate components from these measurements instead of the figure 3 plot.

Figure 2/3 - Did you measure the emergence velocity? I would expect emergence, and this will affect the stress state.

Line 89 - This paragraph seems to shift to modeling methods, from drilling methods.

While it mostly reads ok, perhaps make this a different section? Especially since the section is titled "field site and data acquisition" I also think one paragraph describing the model is a bit thin. If you are actually using the model to interpret your data, please describe it more rigorously and explain the weaknesses with the model output - how much do you trust the modeled principal strain rates and directions? Given that you just assumed a rate factor from another glacier and tuned the sliding to fit this glacier site? Did you conduct a sensitivity study to assess the impact of your parameter selection on the stress and strain rate output from the model? Given that the model inputs are approximate, I'm not entirely sure that the model provides any better qualitative assessment of the expected principal stresses and strain rates than a simpler flow band description explained with clear assumptions.

Line 91 - delete "simply"; say "steady state" model or something like that.

Line 97 - I realize that the model is not intended to be a perfect match, but tuning the model to only one surface velocity is limiting. But I think that's ok, if you are mostly going for the style of stresses and not the real magnitudes (but see my comment above about just using a simpler flow band description because models like this not tuned well can induce complexities that might be interpreted as real). Importantly, the stress distribution with depth at the site of the borehole is highly dependent on the sliding coefficient. When you use these results to interpret the data, please discuss this with respect to the limitations of the model (see my comment above about the vertical distribution of stresses). Oh - and what was used for accumulation/ablation rates? The vertical strain rate at the core site will depend on the ablation rate. Did you measure the vertical velocity at the surface?

Line 123 - delete "as discussed later" and "important" - they don't provide any useful information here.

Figure 5 - nice figure, I am interested in the other 2 eigenvectors - are they equal? Also, please provide some examples of the size distribution (histogram? or statistical distribution curve? You have some statistics in table 2, at a minimum, provide the median. But I would suggest putting the size distribution for each depth in the supplementary information. Put a reference in Figure 2 caption to Table 2 and the supplementary information for the size distribution.

Line 166: "oriented in the direction of glacier flow - just be more specific with wording here. The c-axes points within xxdeg of the flow direction (155deg).

Line 180 - I think this section would be best started with an overview of the deviatoric stress state (if it isn't already in the background), as measured from the surface stakes and as inferred from the model, in terms of the stress tensors and principal directions.

Line 181 - This first sentence doesn't add anything and isn't necessary and is subjective. Just cut it.

Line 192 - see my note from line 180. It is difficult for me to separate the effect of the longitudinal compression alone - I'd rather see a description of the full deviatoric stress state as a function of depth and then look at what components are doing most of the work. Also, there are two horizontal stresses (sigma xx and sigma yy) better to describe these as longitudinal and transverse.

Line 198-202 - Misorientation is most likely, is there a need to go into complex (and incorrect) explanation about surface stress? See my comment above that the "absolute stress" doesn't affect the fabric, only the deviatoric stresses do. This is a really fundamental point, please interpret your fabric in terms of deviatoric stresses.

Line 211 - I'm not sure I understand this, overburden doesn't generate anything, only gradients in overburden (even better to use formal deviatoric stress terminology).

Line 228. I think the author is referring to recrysllization when they say "these processes" - please note that it is not true that they were "just attributed to temperature" - cummulative strain has always been known to be a key part of the process. Please cite earlier work - maybe back from the 70s or 80s on this rather than suggest that this

is new knowledge?

Line 233 Because normal and shear stresses are the two types of stresses, then the statement that a combination of normal and shear must have been involved to create the fabric is minimally useful. Please provide more specific description.

Line 239 what does it mean for a tensor to provide "hints" (that seems to me like an anthropomorphism)?

Line 239/240 Do you mean that this site is not 100% sliding? That's the only way to avoid borehole shearing. It seems like the model set up already defined a limited amount of sliding, there must be some non-zero component of tau xz. So that was an input to the model, not an output.

Line 241 A parabola is typically for an $x^2$ relationship, that is not the case for the curve resulting from Glen's flow law.

Line 243 - how long ago was "recently" can you provide estimates for the timescale of the last significant change in stress state and express that timescale as a percent strain the crystal experiences?

Line 244 - I think the authors mean "latter" not "later"

Line 244 - I'm not quite sure why a mean grain size reduction would necessarily occur after a change in flow direction, unless you are suggesting that the change in direction is triggering specific recrystallization (migration or rotation). I am also not sure I understand the citation to Faria here, as recrystallization has been described in many papers before. Perhaps you can be more specific about what Faria contributed that is specific to this analysis? And please more carefully cite the statements here (alternativly, if you write an overview of the stress state and metamorphism of the crystals in the beginning that describes and cites each process as background and properly cited, you can avoid having to add too many citations in this discussion section.

Line 256 I really like the images of the bubbles and the grain boundaries - it does show

fast grain boundary migration and active interaction between bubbles and boundary movement. How do you know it was a "complete" recrystallization?

Line 258 The image of bubbly and bubble free ice brings up a question I have as to whether there are signs of refrozen water in these thin sections. Water filled crevasses refreeze with a different microstructure that is typically bubble-free or with patterns of bubbles and distinctly different crystals. Some of the small grain "fracture" noted in the paper also might be a post-depositional process. Perhaps it is ok to include these in your analyses, as the same crystal evolution processes are happening, but it might be useful to discuss the ice from snow compaction versus any refrozen water and how that might influence the fabric and grain size distribution (and bubble)

Line 265 - again, I believe you mean to use the word latter.

Line 265 - please define "fast" - fast compared to what? How fast is fast?

Line 273 - delete "as employed in our study?

Line 278 - again, Faria is not the first one to say that temperature is not the only driving process behind boundary migration recrystallization.

Line 280 - be careful using such a strong word as "only" - also, this is a long run-on sentence and would be better to be split up and explained in more specific wording.

Line 280-295 - These sentences don't actually explain how the diamond shape forms, just that it happens at high strain rates in certain orientation of stress. Rewrite this to explain the underlying physical process, if possible. If not possible to explain the physics, then explain this as being associated with specific conditions, with physics still to be determined.

Line 290-291 - The word "only" is too strong, this sentence seems to be a hypothesis you are trying to suggest that your data support (but I don't know what the "certain strain rate" is).

[Figure]

Line 293 - "the absolute strain rate... is expected to be" - please clarify which components of the strain rate tensor you are referring to, or if you mean the effective strain rate (tensor invariant). There is very little discussion of inherited fabric in this paper, How does inherited fabric affect the deeper layers (I don't agree that the surface is necessarily inherited because of any less strain rate at the surface - the only component of the strain rate tensor that is smaller at the surface is the simple shear parallel to the bed).

Line 310 - how do you judge "good agreement"?

Line 317 - "clearly" is not a helpful word - at this point, I am a bit bogged down in generalities and imprecise wording in the fabric and stress/strain relationship, that I am struggling to judge for myself what the source of the 4 maxima are.

Line 335 - I do believe twinning has been observed use EBSD (such as Obbard's work on the Fremont glacier and/or at Siple Dome), I can't remember which one she noted the a-axis alignment that would suggest twinning.

Line 340 - This statement isn't correct, at least the way I am understanding it (increasing overburden/pressure). Please describe the fabric in terms of deviatoric stress tensor as a function of depth, and, in addition, explain more clearly why 4 single maxima are created rather than a girdle, I think you tried to explain this, but it didn't come through very clearly.

Line 346 - yes, in terms of "comprehensive" analysis of the thin sections measurements - this paper is awesome. In terms of interpretation based on stress state, this paper needs work. There have been some other work on temperate glaciers (including some ongoing work on a glacier in Alaska I think - by Gerbi and others? I'm not sure the status of their publications).

---

## Author Comment (AC1) · 18 Sep 2020

Dear Peter Hudleston,

thank you for this correction.

Regards,

Sebastian Hellmann and co-authors
* * *

---

## Author Comment (AC2) · 18 Sep 2020

Dear Peter Hudleston,

We appreciate your constructive and valuable comments to our manuscript tc-2020-133 entitled "Crystallographic analysis of temperate ice on Rhonegletscher, Swiss Alps". We attach a supplementary file with our answers to your specific comments.

We have considered your typographic recommendations and have provided a point-by-point response to your review comments.

If there are further questions, we are happy to answer them and look forward to hearing back from you regarding your decision.

[Figure]

Kind regards,

Sebastian Hellmann and the co-authors.

Please also note the supplement to this comment:
https://tc.copernicus.org/preprints/tc-2020-133/tc-2020-133-AC2-supplement.pdf

**Supplement:**

Dear Peter Hudleston,

We appreciate your constructive and valuable comments to our manuscript tc-2020-133 entitled "Crystallographic analysis of temperate ice on Rhonegletscher, Swiss Alps".
We have considered your typographic recommendations and have provided a point-by-point response to your review comments.

If there are further questions, we are happy to answer them and look forward to hearing back from you regarding your decision.

Kind regards,

Sebastian Hellmann and the co-authors.

*This paper provides a detailed description and analysis of the crystallographic fabric of ice taken from a core from the surface to bedrock in the central part of the ablation zone of a temperate valley glacier. It finds that multimaxima fabrics of the type commonly found in most valley glaciers, usually just from near-surface samples, occur at all depths within the glacier, with some systematic changes with depth in orientation of the clusters that constitute the fabric. This is a new finding and deserves to be published on this basis alone. The paper then, importantly, relates the fabric to the stress field derived from numerical modeling and finds a direct relationship between the orientation of the fabric and orientation of the modeled principal stresses. This leads to a possible explanation of these four maxima fabrics. I question parts of the interpretation and don't believe these fabrics are yet fully explained, as discussed in the specific comments below, keyed to lines in the text. I have also corrected a few typographical errors and made some suggestions for language usage.*

We considered the typographic recommendations in the most recent version.
Based on the reviewer comments, we revised the modelling part and recalculate the values from the model. The new results slightly change our interpretation and also fit better to your explanations. We are going to add the actually derived values for the stress components to the interpretation part to improve the argumentation. Furthermore, we will remove Fig. 9 as it may not fit to the improved results anymore.
The recommendations about grammar and language, especially in the first sections are already included (comments like "changed")

Specific suggestions:

*Line 9-10. The language here doesn't clearly describe the observed relations, since there are four azimuths and colatitudes that define the fabric and three principal stress directions. It is the centroid of the fabric and the maximum principal stress direction that nearly coincide in orientation.*

We changed this sentence:

The centroid of the four-maxima patterns of the individual core samples and the coinciding maximum eigenvector align with the compressive stress directions obtained from numerical modelling.

*Line 31. The stress and kinematic conditions in valley glaciers are more complex than just combinations of simple and pure shear.*

Changed to:

In contrast, for ice samples from temperate glaciers, the deformation is dominated by a series of interfering and variable compressional, extensional, and shear stress conditions along the flow in the valley.

*Line 94. Although the details of the numerical model need not be given here, the basic form of the flow law should be given, since the value of the flow law parameter A is defined. The value of the power law exponent, n, in the flow law should also be noted.*

We revised the complete paragraph as a new section and added some more details according to your and the second reviewer's comments. Furthermore, we incorporated most recent data about bedrock sliding and tuned the model accordingly.

*Line 117. It is not clear what is meant by fractures here, since there are no actual fractures in this*

*core. This needs clarification. What are the physical manifestation of the 'fractures?" They must be defined by some combination and bubble or grain size distribution.*

We changed it to "fracture traces" as recommended in your comment for Line 151. However, in other literature we found the term "fissures".

*Line 135. Surely this is mm2 not µm2*

Indeed, this must be mm2, changed.

*Line 151. Here is some information about the fractures. Presumably these patterns are in the form of linear traces in thin section. Following Hambrey I like the term 'fracture traces' for these likely healed fractures.*

We changed it to "fracture traces" as a much better name for these features that could be observed in some core depths.

*Line 157. You use the term centroid here for the maximum eigenvector on these plots, and state that these are equivalent in the caption to Table 1. Yet in Figure 7 the two are represented and plotted as separate entities. The usage needs to be consistent. In this case how is centroid defined?*

We revised the usage of centroid and centre (i.e. midpoint) between the clusters. The midpoint (red dots in Fig. 7) is defined as geometric point between the four maxima (independent of number of grains per cluster). The centroid is affected by the particular distribution of grains and those maxima with a larger grain number attract the centroid. Therefore, midpoint and centroid differ slightly. When calculating the opening angle we considered the midpoint as symmetry point of the multi-maxima pattern.

*Line 173-174. It should be noted that Kamb, Hooke and others have discussed the issue of accounting for complex and branching shapes of large grains when making c-axis plots.*

We added the recommended references and furthermore two papers that also show images for a better visualisation:

Therefore, two-dimensional cuts through large, branched grains may let them appear as several individual grains within the same section. Kamb (1959) and Hooke (1969) have already discussed the statistical relevance of these branched grains. The sketches in Hooke (1980), Fig. 6 and more recently in Monz (2020), Fig. 3, further illustrate this issue that could result in over-represented clusters in the superimposed stereo plots from the different sub-samples.

*Fig. 6. The caption could be shortened by stating that the annotation is as in Fig. 5*

Changed.

*Line 192. The c-axis fabric has orthorhombic (and perhaps close to axial) symmetry, but is this also true of the stresses? What about the other two principal stresses. Are they consistent with plane strain or plane stress, as appears to be assumed in Fig. 9? Are the principal stresses and strain rates in this section of the glacier near the surface parallel to the flow direction ($\sigma$1), vertical ($\sigma$ 3) and horizontal ($\sigma$2), with the lateral strain rate close to zero, as one would expect for a valley of constant width. One would expect the maximum principal stress to become inclined deeper into the ice as shear stress parallel to the base increases, which the modeled stress shows a tendency to do.*

We will add the other two principal components in Fig. 7. Although the lateral strain rate is small, it is not zero, neither as result of the model nor from our borehole displacements at the surface. The modelled principal stresses are not perfectly aligned with the calculated eigenvectors from our COF analysis. They show a slight offset of 25°. However, the eigenvectors (blue dots in Fig 9) are the actually calculated directions of the principal axes derived from our measured COF distribution and not only an assumption. In all other depths, the distribution of the eigenvectors is similar.

*Line 200-204. This is a possible explanation, but I prefer the misorientation of the sample as the explanation, which as you state, fits very nicely when a 60o azimuthal 'correction' is made. A preserved fabric from earlier in the flow path is less likely at high temperatures when rapid recrystallization is expected.*

We removed this immature hypothesis, especially as we cannot see a smaller deviatoric stress in the uppermost parts.

*Lines 210-213. With this explanation you would expect σ1 to be vertical to explain the fabric at 79m depth and not as given by the numerical model. Although the vertical normal stress increases with depth, it is the deviatoric stress that controls deformation, not absolute stress values, and this likely does not change greatly with depth. I think the main thing that changes with depth is not the vertical effective compression (σ1 – σmean) but the increasing addition of base parallel shear stress, that in general terms increases linearly with depth.*

Our mistake was to consider a (hydrostatic) overburden pressure. However, this hydrostatic pressure does not contribute to the deviatoric stress that drives the c-axis orientation (via strain rates). We revised this part accordingly. As you say, the c-axis orientation (i.e. the centroid) in the deepest part is in alignment with shear stress: Base-parallel shear stress lead to a base-parallel orientation of the basal planes and thus a more vertical c-axis. The model shows that the shear stress component $\sigma^d_{xz}$ (which we will add to the results section) is the most dominant stress and the in-flow compressional component $\sigma^d_{xx}$ is much smaller in this depth (similar to $\sigma^d_{yy}$).

*Line 217. There is almost certainly some dependence of fabric on strain, which may not be great with fast recrystallization.*

Here you suggest to also consider the strain (and not strain rates)? Indeed, recrystallization may appear only if the strain load is high enough. However, we need to analyse, if an immediate (and complete, i.e. all grains recrystallize) recrystallization is necessary or if we could also have a recrystallization of some grains that allow a strain reduction for the whole polycrystal and then other grains with less suitable orientations can still exist until the strain becomes too large and they recrystallize too.

*Line 213-214. In simple shear the directions of principal stress are only aligned with those of principal strain for infinitesimal strains. The divergence grows as strain increases.*

We assume that strain rate (and the strain) and stress direction form an angle of ~45°. Therefore, the principal stress direction (governed by the simple shear component) in the deepest part of ~50° and the actual centroid (~0°) would fit under such an assumption. The MM cluster is aligned with the strain rate direction in that depth and not with stress as stress and strain diverge for simple shear.

*Line 238 I don't believe Cuffey and Paterson really explain why there should be four maxima when the stress deviates from unconfined compression. This is more of an observation than an explanation.*

No, they only provide a description and mention different stresses are required for multi-maxima. We considered this in the revised manuscript.

*Lines 244-245. This is unclear. A change in direction of glacier flow could be associated with either an increase or decrease in strain rate and thus decrease or increase in recrystallized grain size. Why just a decrease?*

This is correct, it could be an increase as well. However, here we want to highlight the effect of strain-induced boundary migration with new nuclei (SIBM-N). In this particular type of recrystallization, grain growth counteracts against recrystallization with new and usually smaller grains. We need to rethink how we can make this clear.

*Line 251 and Table 2. Table 2 does not really give the grain size distribution, only average numbers of grains and average and maximum size in each sample. It would be useful to know the number of grains in each size category. Also interesting to know if there is any difference between the large and small grains in COF.*

We added a supplementary figure (according to the comments of the second reviewer) and median and 6 different grain size classes to Table 2.

*Line 255-266. I'm not sure how much information is given by the air bubbles, except they do provide excellent evidence of active recrystallization by grain boundary migration. Bubbles are found both within grains and along grain boundaries both in temperate ice and in cold ice experiencing dynamic recrystallization, although the recrystallization mechanism may differ.*

As you said, we want to provide an example for recrystallization here. However, we expect to find more air bubbles along grain boundaries if the recrystallization takes place much slower. Air bubbles could be seen as obstacles and thus more energy (from strain rate or temperature) is required to fully incorporate these obstacles in the grain matrix rather than having an accumulation along dislocations.

*Line 269. Hooke and Hudleston were concerned with polar, not temperate ice. The study was made on the Barnes Ice Cap.*

Thank you, again a valuable hint. We will change it:
They were observed in early studies on temperate glaciers (e.g., Rigsby, 1951; Kamb, 1959; Rigsby, 1960), ice capes with ice temperatures above -10°C (Hooke and Hudleston, 1980), and also in the bottom ice of Byrd Station and Cape Folger in Antarctica (Gow and Williamson, 1976; Thwaites et al., 1984). They are often referred as "diamond-shape" pattern or fabrics.

*Line 276. Whether the multimaxima fabrics are a result of unrepresentative sampling is still arguable in some circumstances, although the case you have here for these being true multimaxima fabrics is a strong one.*

Here, the additional analysis according to your comment in Line 251/Table 2 may be useful to show that there are multi-maxima, but especially for the smallest grains, which may result from recrystallization, the analysis shows a preference for selected clusters.

*Line 290-291. I think more data is needed to support this conclusion. The cores taken by Tison and Hubbard were in a different regime within the glacier – accumulation zone where perhaps there is*

*longitudinal extension rather than compression, and close to the lateral margin of the glacier rather than in the center. This must lead to a more complex stress regime.*

Actually, some cores were drilled in the ablation zone and in these cores they found multi-maxima at the bottom. However, it is true that they are drilled at the margin and therefore they could significantly differ from our core close to the centre flow line of the glacier. This could complicate a direct comparison.

*Line 298-299. The combination of compression plus simple shear as applied in these experiments makes sense for much of your core, but not for the highest one where the shear component is minimal, nor for the lowest one, where the σ1 direction lies well outside the small-circle girdle of maxima. Some other explanation must hold in these places.*

In case of a misorientation in the upper part, our explanation would fit as well. However, even after recalculating the deviatoric stress, we cannot find a proper solution for the depth of 79 m. The diamond pattern seems to cluster around the direction of the strain rate induced by shear stress only (if assuming that shear stress and horizontal compression are the two dominant deviatoric stresses).

*Line 300-301. I'm not sure if I'm properly interpreting what you are saying here, but the planes of maximum shearing stress in Duval's combined compression-simple shear experiments are not vertical and horizontal in his experiments, but inclined by an amount that depends on the relative amounts of normal compression and simple shear.*

This is exactly, what we wanted to cite here. His experiments show a multi-maximum pattern that is aligned with the compressional axis and two of these maxima are also aligned with the poles of the two shear planes as the angle between the maxima and the principal direction is roughly

*Line 312-316. Both Llorens et al. and Qi et a. are dealing only with simple shear, not with combined compression plus simple shear as in the torsion plus compression experiments. The conditions in the Rhone glacier I imagine change from horizontal compression with minimal base-parallel shear near the surface to horizontal compression combined with increasing base-parallel shear near the base of the glacier. As theory shows, shear stress increases approximately linearly with depth, while longitudinal stress stays approximately constant.*

The horizontal compression slightly decreases and the base-parallel shear stress significantly increases with depth. With these modelling results in Llorens et al and Qi et al, we wanted to discuss whether some of the maxima can be a result of the shear stress. The idea was to separate the maxima and explain how they develop as a result of different crystals that react as a network against different stress conditions. We have to admit that this was very speculative and needs to be revised. However, these localisation effects, i.e. some crystals recrystallize immediately to act against certain strain rates whereas other grains could still keep their orientation is a point that we reformulate in our improved version but with less emphasis as we tried to do in the first draft.

*Fig. 9. The stress state shown in Fig. 9 is almost that of simple shear (no base-parallel longitudinal compression) with the shear plane (taken as the glacier bed) horizontal and σ1 inclined at 45o to the shear plane. If it is simple shear, there will be no horizontal compression and thus no shortening in the glacier flow direction, which is incompatible with your data. If horizontal glacier flow-parallel compression is added σ1 will move closer to horizontal than it would be for simple shear alone. This looks like being the case for much of the glacier from the stresses shown in Fig. 7. I would expect the inclination of σ1 to be near zero at the surface and something less than 45o close to bedrock, the amount depending on the amount of horizontal compression. Although not a smooth change, the σ1*

*directions in Fig. 7 are consistent with this. The plot in Fig. 9 does not correspond to any of the plots in Fig. 7, all of which have σ1 at a shallower inclination than 45o and thus have associated planes of maximum shearing stress that are neither vertical or horizontal, unlike the situation in Fig. 9. The one closest to horizontal thus cannot be considered a plane of simple shear. The 'shear plane' must always be the presumably sub-horizontal glacier bed.*

Same as our comment for line 312-316, this figure was planned to support our assumption that particular clusters react against the different stresses.

*Line 340. I disagree with the statement here (see comments for lines 210-213). Although the absolute value of the vertical normal stress increases with depth, the deviatoric vertical normal stress changes much less. It is the increase in base-parallel shear stress combined with the horizontal compressive stress (σxx if you like) that causes σ1 to rotate from near horizontal at the surface to inclined at some angle of less than 45° at the base.*

Indeed, the vertical stress $\sigma^d_{zz}$ is not responsible and actually decreases in our revised model with increasing depth. We will change this conclusion accordingly and agree with your suggestion that the orientation is driven by $\sigma^d_{xz}$ and $\sigma^d_{xx}$

*Line 342. The second part (ii) of the explanation for multimaxima fabrics given here makes no sense by itself. All states of stress that are non isotropic involve shear stresses. If the multimaxima fabric depended solely on the state of stress – that is with instantaneous adjustment of the c-axis fabric as the stress field changes – then there should be a constant relationship between the positions of the maxima and the principal stress directions. This clearly is not the case as the relationship in the deepest sample shows. There is, however, as you note, a consistent relationship between the fabric and the σ1 direction through most of the glacier and in all cases, with small deviations, the centroid of the COF fabric and the σ1 direction lie in the vertical plane that contains the flow direction. This is a key relationship that I believe you have only partly explained.*

We are going to revise the interpretation and discussion. Then, we also put a larger emphasis on the fact that the general COF pattern is aligned with the flow direction (with an exception in 79 m).

---

## Author Response (AR1)

Dear Erin Pettit,

Thank you very much for your valuable comments. We appreciate your constructive feedback, which helped to enhance the quality of our manuscript tc-2020-133 entitled "Crystallographic analysis of temperate ice on Rhonegletscher, Swiss Alps".
We have considered your suggestions for grammar and writing style and have provided a point-by-point response to your review comments.

If there are further questions, we are happy to answer them and look forward to hearing back from you regarding your decision.

Kind regards,

Sebastian Hellmann and the co-authors.

**General comments**

*This paper provides a careful analysis of the measured fabric in the central part of a Rhonegletscher, in the ablation area through ice cores (not quite to bedrock).*
*The authors did a really thorough and rigorous analysis of the fabric using multiple thin sections in 3 orientations. The most well analyzed core I have seen for a temperate glacier, I appreciated the thoroughness, as it was necessary because of the dominance by large grains and a grain size distribution that if far from normal. This paper in some form should be published because of the beautiful data set.*

*While the data analysis is done really well, the interpretation in terms of stress state is not as thorough and rigorous. Their qualitative interpretation of the stress state and its relation to fabric and recrystallization processes is confusing and in a few places incorrect. The paper would benefit from a summary of the key states of stress, key metamorphic processes, citing the original research (going beyond Cuffey and Paterson and the Faria reviews). As a reader, if I am to trust their interpretation of the fabric, I need to trust that they understand the underlying physics. At this point, the physics is description is still lacking. It is imperative that the interpretation of the fabric in terms of the stresses be written with the same care and rigor that that fabric was measured.*

*First, it would be helpful to clarify when deviatoric stress is being used versus total stress. Deviatoric stress control most of the deformation and pressure plays a minor, if any, role in deformation, therefore describe the deviatoric stress states rather than "absolute" stress and "overburden." For example, the authors suggest that there is less deformation in the surface layers because the "absolute" stress there is low, but this is not the case - the vertical compressive deviatoric stress is no necessarily smaller at the surface, it is typically about the same - it is only the pressure term in the total stress that is smaller at the surface and pressure does not drive fabric (only gradients in pressure or overburden can drive flow).*
*I would suggest that they re-write the description of the stress state in terms of more formal tensor components, and more specific (and correct) wording. And please be explicit about what is behavior linked to stress and what is behavior linked to strain rate (and discuss with respect to the statement that strain rate ultimately drives fabric development not stress).*
*In terms of writing, there are numerous run-on sentence, imprecise wording, and extensive use of passive voice, all of which slows down the reader. I provide examples of a few of these (but not all of them) below, I encourage the authors to edit carefully for these three writing issues.*

We will check for any passive structures and run-on sentences and appreciate your particular recommendations below. In the revised version, we introduce a subsection about the physical details. We also show the deviatoric stresses for each depth in a separate table and use the particular values for an improved discussion. Of course, the overburden pressure is not responsible for deformation of the ice. We have rectified this blunder.

**Specific suggestions**

*Line 1 Abstract - the first line of the abstract should offer some kind of bigger picture motivation, something to entice readers beyond those already rheology and fabric "geeks" - this is a neat paper with respect to the unique measurements and it would nice for the broader glaciology community to read it.*

We added a more general introductory sentence:
The crystal orientation fabrics (COF) provide key information about the mechanics of glacier flow as its development is driven by a combination of stresses, strain and recrystallisation. Detailed information of COF can be considered to improve specific parameters for glacier modelling.

*Line 20: delete "to that" - not necessary*

Removed

*Line 23: delete "do"*

Removed

*Line 24: Faria offered great reviews in his 2014 papers, but be careful citing those*
*papers when there are better papers that are more directly or more originally related*
*to the statement. Here by citing Faria, it implies that that paper was the first to discover that COF evolves in a glacier. Provide more direct/original citations please (or be*
*explicit that you are citing Faria as a review article).*

We added the original work:
The stresses and strains occurring within the ice mass not only cause glacier flow, but also induce the development of a characteristic COF and microstructural anisotropy (Gow and Williamson, 1976; Herron and Langway, 1982; Alley et al., 1995, 1997) and summarised in Faria et al. (2014a).

*Line 30: delete "quickly" unless you want to provide the timescale that quickly is indicative of (words like quickly, clearly, mostly, etc don't add any information and can lead to*
*confusion).*

Removed, we revised this sentence according to the first reviewer's comment.

*Line 53 - run-on sentence, breakup into two or three.*

We changed it as follows:

To date, ice core drilling and preparation of thin sections is still a time-consuming process. Only a few discrete measurements are possible within a reasonable amount of time. Nonetheless, the technique for analysing COF has developed extensively, for example, by using image analysis software and powerful

computing resources (Wilson et al., 2003; Peternell et al., 2009; Wilson and Peternell, 2011; Eichler, 2013).

*Lines 74-78 - lots of passive voice here, rewrite*

We changed this to a more active style:

As the ice is just at the pressure melting point, we used a thermal drilling technique (Schwikowski et al., 2014). Although hot-water drillings, performed in the vicinity of the ice core location, showed a mean ice thickness of 110 m, we stopped drilling at 80 m, when hitting some gravel. This gravel blocked the cutter head. We retrieved an 80 m long ice core, with a gap between 46 and 50 m due to technical issues.

*Line 78 - I'm still a little confused how you knew the orientation, did the drill head not*
*spin on the cable as it was lowered or raised?*

Indeed, the core barrel and drill head could rotate, but a magnetometer was integrated into this core barrel. After each drilling we turned this core barrel until reaching the orientation at beginning. Then the core segment was retrieved and its orientation was marked with a knife. Afterwards we tried to attach the segment to the previous one. If this was possible we added a notch with a soldering iron on both segments. However, if this was not possible, we opened up the notch of the knife. Later, we could retrieve the orientation of the core barrel during the drilling process. Assuming that the core segment was not rotating within the core barrel (e.g. due to sudden shocks which we avoided by a decent winch speed), we could retrieve the actual core orientation.
The data also reveal that there is no 360°-spinning around the cable (just slight movements). The water-filled borehole damped any rotation of the core barrel.

*Figure 2/3: I like the diagram in figure 3, but why not just calculate the bulk surface*
*strain rate components from these measurements instead of the figure 3 plot.*

We actually calculate the strain rates as constraint for our model. We will add this information and discuss whether it could replace Fig 3 (or if we keep both as the figure emphasises the smooth decrease in flow speed along ice flow).

*Figure 2/3 - Did you measure the emergence velocity? I would expect emergence, and*
*this will affect the stress state.*

We did not measure this value at this position. However, a reference station about 50 m away from the boreholes shows an emergence of 1.5-2 m $a^{-1}$.

*Line 89 - This paragraph seems to shift to modeling methods, from drilling methods.*
*While it mostly reads ok, perhaps make this a different section? Especially since the*
*section is titled "field site and data acquisition" I also think one paragraph describing*

*the model is a bit thin. If you are actually using the model to interpret your data, please*
*describe it more rigorously and explain the weaknesses with the model output - how*
*much do you trust the modeled principal strain rates and directions? Given that you just*
*assumed a rate factor from another glacier and tuned the sliding to fit this glacier site?*
*Did you conduct a sensitivity study to assess the impact of your parameter selection*
*on the stress and strain rate output from the model? Given that the model inputs are approximate, I'm not entirely sure that the model provides any better qualitative*
*assessment of the expected principal stresses and strain rates than a simpler flow*
*band description explained with clear assumptions.*

We improved the modelling description and moved it to a new section (see substantial changes).
According to your questions: As you point out in your comment for line 97, the model is only used to constrain our interpretation. However, we could have used a flow band description or our borehole data to interpret. The main task behind employing the model is to get some quantitative values rather than just speculating qualitatively about different stresses (namely compressional in-flow and shear stress). The most important weakness is that we use the stress information of a single point to explain the stress conditions for a certain area of the glacier. Local stress effects cannot be captured by such a model. Furthermore, we also do not have reliable bed velocities and we have to admit that the model is only constrained by surface velocity and ice thickness information. To overcome these weaknesses, we will consider strain rates derived from our borehole experiments (Table 3).

*Line 91 - delete "simply"; say "steady state" model or something like that.*

We revised the whole paragraph and removed it.

*Line 97 - I realize that the model is not intended to be a perfect match, but tuning the*
*model to only one surface velocity is limiting. But I think that's ok, if you are mostly going for the style of stresses and not the real magnitudes (but see my comment above*
*about just using a simpler flow band description because models like this not tuned*
*well can induce complexities that might be interpreted as real). Importantly, the stress*
*distribution with depth at the site of the borehole is highly dependent on the sliding*
*coefficient. When you use these results to interpret the data, please discuss this with*
*respect to the limitations of the model (see my comment above about the*

*vertical distribution of stresses). Oh - and what was used for accumulation/ablation rates? The*
*vertical strain rate at the core site will depend on the ablation rate. Did you measure*
*the vertical velocity at the surface?*

We do not model a time-transient evolution but only calculate the stress field for the actual geometry. Therefore, we did not consider the accumulation and ablation rates. The limitation is that the model only provide stresses and not strain rates or directions. We can calculate strain rates with Glen's flow law. Please regard the model as constraining information for the interpretation. We do not intend to setup a perfect model that describes the ice flow and stress + strain rates. This needs additional measurements and is beyond the scope of this work. In the revised interpretation, we also consider strain rates derived from the borehole data.

*Line 123 - delete "as discussed later" and "important" - they don't provide any useful*
*information here.*

Changed.

*Figure 5 - nice figure, I am interested in the other 2 eigenvectors - are they equal? Also,*
*please provide some examples of the size distribution (histogram? or statistical distribution curve? You have some statistics in table 2, at a minimum, provide the median.*
*But I would suggest putting the size distribution for each depth in the supplementary*
*information. Put a reference in Figure 2 caption to Table 2 and the supplementary*
*information for the size distribution.*

We add the other two eigenvectors (Fig 5), symbol size decreases accordingly. The eigenvalues are not 100% equal but both around 0.10-0.31 (we added the particular ranges in the text, line 181). Usually the second eigenvector is laying in the vertical plane of the diamond shape pattern.
We also add the number of grains for 6 grain size classes (<1/1-5/5-20/20-100/100-500/>500 mm$^2$) to Table 2 and show a histogram for a selected depth and put the others to the supplement. We will use this additional figure in our interpretation as the small grains (<1mm$^2$) clearly emphasis one of the four clusters. This provides some evidence that recently recrystallized grains in the deeper and intermediate parts of the glacier prefer one of the clusters rather than equally distribute to all four clusters.

*Line 166: "oriented in the direction of glacier flow - just be more specific with wording here. The c-axes points within xxdeg of the flow direction (155deg).*

We changed it as follows:

… the azimuth of the maximum eigenvectors (147° ± 31°) is aligned with the direction of the glacier's ice flow (155◦ ± 10◦, cf. Figs. 1, 2).

*Line 180 - I think this section would be best started with an overview of the deviatoric stress state (if it isn't already in the background), as measured from the surface stakes and as inferred from the model, in terms of the stress tensors and principal directions.*

See substantial changes: we introduce the interpretation with such an overview in lines 200-218. In addition, we add tables with the respective components of stress and strain rate tensor.

*Line 181 - This first sentence doesn't add anything and isn't necessary and is subjective. Just cut it.*

Removed.

*Line 192 - see my note from line 180. It is difficult for me to separate the effect of the longitudinal compression alone - I'd rather see a description of the full deviatoric stress state as a function of depth and then look at what components are doing most of the work. Also, there are two horizontal stresses (sigma xx and sigma yy) better to describe these as longitudinal and transverse.*

See substantial changes: we introduce the interpretation with such an overview in lines 200-218. In addition, we add tables 2+3 with the respective components of stress and strain rate tensor.

We also considered your recommendation to distinguish between the dominant longitudinal and the transversal stress.

*Line 198-202 - Misorientation is most likely, is there a need to go into complex (and incorrect) explanation about surface stress? See my comment above that the "absolute stress" doesn't affect the fabric, only the deviatoric stresses do. This is a really fundamental point, please interpret your fabric in terms of deviatoric stresses.*

We removed this immature argumentation.

*Line 211 - I'm not sure I understand this, overburden doesn't generate anything, only gradients in overburden (even better to use formal deviatoric stress terminology).*

We completely revised this paragraph (see substantial changes). The parts where we referred to overburden pressure have been removed. Instead, we introduce our interpretation with an explanation that it is the deviatoric stress driving the deformation.

*Line 228. I think the author is referring to recrysllization when they say "these processes" - please note that it is not true that they were "just attributed to temperature" - cummulative strain has always been known to be a key part of the process. Please cite earlier work - maybe back from the 70s or 80s on this rather than suggest that this is new knowledge?*

Here (and more in detail in the discussion), we introduce the concept of Faria et al. and also show, where their concept differs from previous literature (see substantial changes). Faria clearly state that the tripartite paradigm is wrong and our interpretation is based on their assumptions (which has been proven by other authors in the last years).
They particularly distinguished between strain-induced boundary migration with new grains (SIBM-N) and strain-induced boundary migration with keeping the old grains (SIBM-O).

*Line 233 Because normal and shear stresses are the two types of stresses, then the statement that a combination of normal and shear must have been involved to create the fabric is minimally useful. Please provide more specific description.*

We completely removed this immature part. Instead we refer more to the recrystallization processes to describe the diamond shape pattern.

*Line 239 what does it mean for a tensor to provide "hints" (that seems to me like an anthropomorphism)?*

This line has been removed. See comments to Line 233 and substantial changes.

*Line 239/240 Do you mean that this site is not 100% sliding? That's the only way to avoid borehole shearing. It seems like the model set up already defined a limited amount of sliding, there must be some non-zero component of tau xz. So that was an input to the model, not an output.*

Our model assumes basal sliding (parameter c > 0) and we removed this line during our revision.
However, we also considered no basal sliding in our model for a sensitivity analysis. This would lead to giant rate factors which are unrealistic. Therefore, basal sliding is, indeed, a prerequisite.

*Line 241 A parabola is typically for an x^2 relationship, that is not the case for the curve resulting from Glen's flow law.*

It is a hyperbolic curvature.

*Line 243 - how long ago was "recently" can you provide estimates for the timescale of the last significant change in stress state and express that timescale as a percent strain the crystal experiences?*

"Recently" must be within the last four decades as the ice flow direction changed about 1000 m up-glacier and our pattern is in good agreement with the current flow direction. The strain % is difficult to assess as we do not have information about the surface velocities in that area further up-glacier.

*Line 244 - I think the authors mean "latter" not "later"*

Changed.

*Line 244 - I'm not quite sure why a mean grain size reduction would necessarily*

*occur after a change in flow direction, unless you are suggesting that the change in direction is triggering specific recrystallization (migration or rotation). I am also not sure I understand the citation to Faria here, as recrystallization has been described in many papers before. Perhaps you can be more specific about what Faria contributed that is specific to this analysis? And please more carefully cite the statements here (alternativly, if you write an overview of the stress state and metamorphism of the crystals in the beginning that describes and cites each process as background and properly cited, you can avoid having to add too many citations in this discussion section.*

As described in our answer to line 228, Faria was (to our knowledge) the first, who distinguished between SIBM-O and SIBM-N. Others only referred to dynamic and rotational recrystallization (RRX). This distinction is particularly important for observed grain-size changes at high temperatures as in our case. That even leads to a new process understanding, e.g. Steinbach et al (2017) in Frontiers in Earth Science, Vol 5.
We include a paragraph in the discussion and describe in detail, how the findings of Faria et al. differ from previous studies (see substantial changes).

*Line 256 I really like the images of the bubbles and the grain boundaries - it does show fast grain boundary migration and active interaction between bubbles and boundary movement. How do you know it was a "complete" recrystallization?*

We actually do not need this Figure anymore and do not use the observations in our revised version.

*Line 258 The image of bubbly and bubble free ice brings up a question I have as to whether there are signs of refrozen water in these thin sections. Water filled crevasses refreeze with a different microstructure that is typically bubble-free or with patterns of bubbles and distinctly different crystals. Some of the small grain "fracture" noted in the paper also might be a post-depositional process. Perhaps it is ok to include these in your analyses, as the same crystal evolution processes are happening, but it might be useful to discuss the ice from snow compaction versus any refrozen water and how that might influence the fabric and grain size distribution (and bubble)*

We have seen those fracture traces in two depths (22+45 m). We also analysed these fracture grains separately and can provide information about their orientation. Some of them are perfectly aligned with the surrounding large grains (especially if the fracture is thin). Others (if not a fracture but rather a patch of small grains) show a girdle structure. This girdle is aligned with the glacier flow (extension in transverse direction). We could exclude these grains from our analysis. This would emphasise the diamond pattern in 22 and 45 m.

*Line 265 - again, I believe you mean to use the word latter.*

Changed.

*Line 265 - please define "fast" - fast compared to what? How fast is fast?*

In the revised version, we do not refer to "fast" recrystallization anymore. However, we included a sentence that the orientation of the patter is in

alignment with the current glacier flow. The glacier has flown in this direction for about 30-40 years.

*Line 273 - delete "as employed in our study?*

Changed.

*Line 278 - again, Faria is not the first one to say that temperature is not the only driving process behind boundary migration recrystallization.*

This is true, but to our knowledge, Faria is the first one who distinguished between SIBM-N and SIBM-O which assumes a reorientation of old grains (SIBM-O) or a complete creation of small new grains (SIBM-N). According to his model and considering our strain rates and temperatures we have SIBM-N conditions here. This is the difference to earlier studies.
We rephrased according to the calculated strain rates from borehole measurements.

See substantial changes for more details.

*Line 280 - be careful using such a strong word as "only" - also, this is a long run-on sentence and would be better to be split up and explained in more specific wording.*

Thank you, we rewrote this long sentence and consider, that we do not have 100% evidence for our argumentation and thus "only" is obmitted here.

*Line 280-295 - These sentences don't actually explain how the diamond shape forms, just that it happens at high strain rates in certain orientation of stress. Rewrite this to explain the underlying physical process, if possible. If not possible to explain the physics, then explain this as being associated with specific conditions, with physics still to be determined.*

We cannot explain the exact physical processes but we rewrite our suggestions that may be responsible for the diamond pattern.

*Line 290-291 - The word "only" is too strong, this sentence seems to be a hypothesis you are trying to suggest that your data support (but I don't know what the "certain strain rate" is).*

Here we need to be more conclusive and argue with the actual values of strain rate.

*Line 293 - "the absolute strain rate... is expected to be" - please clarify which components of the strain rate tensor you are referring to, or if you mean the effective strain rate (tensor invariant). There is very little discussion of inherited fabric in this paper, How does inherited fabric affect the deeper layers (I don't agree that the surface is necessarily inherited because of any less strain rate at the surface - the only component of the strain rate tensor that is smaller at the surface is the simple shear parallel to the bed).*

This paragraph was revised. The strain rate and also the deviatoric stress is not smaller at the surface compared to other depth (Table 2).

*Line 310 - how do you judge "good agreement"?*

We wanted to point out that the stress conditions in these laboratory experiments are in a similar range as we find them in the glacier. During our revision, we rephrased this part.

*Line 317 - "clearly" is not a helpful word - at this point, I am a bit bogged down in generalities and imprecise wording in the fabric and stress/strain relationship, that I am struggling to judge for myself what the source of the 4 maxima are.*

We avoid these words in the revised version.

*Line 335 - I do believe twinning has been observed use EBSD (such as Obbard's work on the Fremont glacier and/or at Siple Dome), I can't remember which one she noted the a-axis alignment that would suggest twinning.*

Up to date, we could not find the respective part in the papers of R.W Obbard. However, her work is worth to consider as it clearly points out the ambiguities of our technique (analysing the c-axis without the a-axes information).

*Line 340 - This statement isn't correct, at least the way I am understanding it (increasing overburden/pressure). Please describe the fabric in terms of deviatoric stress tensor as a function of depth, and, in addition, explain more clearly why 4 single maxima are created rather than a girdle, I think you tried to explain this, but it didn't come through very clearly.*

Indeed, this needed a revision. The overburden pressure is hydrostatic and not responsible for strain rates that drive c-axis developments. We removed this argumentation and included a paragraph in which we describe the physics (i.e. deviatoric stresses) leading to deformation and COF changes.

*Line 346 - yes, in terms of "comprehensive" analysis of the thin sections measurements - this paper is awesome. In terms of interpretation based on stress state, this paper needs work. There have been some other work on temperate glaciers (including some ongoing work on a glacier in Alaska I think - by Gerbi and others? I'm not sure the status of their publications).*

We agree that we have to revise the interpretation part and added a couple of additional details about the stress state in the glacier as derived from the model and further provided information about the strain rates from in situ measurements. These data should simplify the interpretation and allow a better access to the information provided in our paper.
We also figured out that there are a couple of presentations from Gerbi at AGU. However, there seems to be no field data published yet.

Dear Peter Hudleston,

We appreciate your constructive and valuable comments to our manuscript tc-2020-133 entitled "Crystallographic analysis of temperate ice on Rhonegletscher, Swiss Alps".
We have considered your typographic recommendations and have provided a point-by-point response to your review comments.

If there are further questions, we are happy to answer them and look forward to hearing back from you regarding your decision.

Kind regards,

Sebastian Hellmann and the co-authors.

*This paper provides a detailed description and analysis of the crystallographic fabric of ice taken from a core from the surface to bedrock in the central part of the ablation zone of a temperate valley glacier. It finds that multimaxima fabrics of the type commonly found in most valley glaciers, usually just from near-surface samples, occur at all depths within the glacier, with some systematic changes with depth in orientation of the clusters that constitute the fabric. This is a new finding and deserves to be published on this basis alone. The paper then, importantly, relates the fabric to the stress field derived from numerical modeling and finds a direct relationship between the orientation of the fabric and orientation of the modeled principal stresses. This leads to a possible explanation of these four maxima fabrics. I question parts of the interpretation and don't believe these fabrics are yet fully explained, as discussed in the specific comments below, keyed to lines in the text. I have also corrected a few typographical errors and made some suggestions for language usage.*

We considered the typographic recommendations in the most recent version. Based on the reviewer comments, we revised the modelling part and recalculate the values from the model. The new results slightly change our interpretation and also fit better to your explanations. We are going to add the actually derived values for the stress components to the interpretation part to improve the argumentation. Furthermore, we will remove Fig. 9 as it may not fit to the improved results anymore.
The recommendations about grammar and language, especially in the first sections are already included (comments like "changed")

**Specific suggestions:**

*Line 9-10. The language here doesn't clearly describe the observed relations, since there are four azimuths and colatitudes that define the fabric and three principal stress directions. It is the centroid of the fabric and the maximum principal stress direction that nearly coincide in orientation.*

We changed this sentence:

The centroid of the four-maxima patterns of the individual core samples and the coinciding maximum eigenvector align with the compressive stress directions obtained from numerical modelling.

*Line 31. The stress and kinematic conditions in valley glaciers are more complex than just combinations of simple and pure shear.*

Changed to:

In contrast, for ice samples from temperate glaciers, the deformation is dominated by a series of interfering and variable compressional, extensional, and shear stress conditions along the flow in the valley.

*Line 94. Although the details of the numerical model need not be given here, the basic form of the flow law should be given, since the value of the flow law parameter A is defined. The value of the power law exponent, n, in the flow law should also be noted.*

See our remarks for substantial changes. We added a new section and provide the basic equations for ice flow and the Weertman's friction law. We also define/describe the respective parameters.

*Line 117. It is not clear what is meant by fractures here, since there are no actual fractures in this core. This needs clarification. What are the physical manifestation of the 'fractures?" They must be defined by some combination and bubble or grain size distribution.*

We changed it to "fracture traces" as recommended in your comment for Line 151. However, in other literature we found the term "fissures". As it seems not to be conclusive, we use both terms.

*Line 135. Surely this is mm2 not $\theta$m2*

Indeed, this must be mm2, changed.

*Line 151. Here is some information about the fractures. Presumably these patterns are in the form of linear traces in thin section. Following Hambrey I like the term 'fracture traces' for these likely healed fractures.*

We changed it to "fracture traces" as a much better name for these features that could be observed in some core depths. However, they are sometimes called fissures. Therefore, we mention both names.

*Line 157. You use the term centroid here for the maximum eigenvector on these plots, and state that these are equivalent in the caption to Table 1. Yet in Figure 7 the two are represented and plotted as separate entities. The usage needs to be consistent. In this case how is centroid defined?*

We revised the usage of centroid and centre (i.e. midpoint) between the clusters. The midpoint (red dots in Fig. 7) is defined as geometric point between the four maxima (independent of number of grains per cluster). The centroid is affected by the particular distribution of grains and those maxima with a larger grain number attract the centroid. Therefore, midpoint and centroid differ slightly. When calculating the opening angle we considered the midpoint as symmetry point of the multi-maxima pattern.

*Line 173-174. It should be noted that Kamb, Hooke and others have discussed the issue of accounting for complex and branching shapes of large grains when making c-axis plots.*

We added the recommended references and furthermore two papers that also show images for a better visualisation:

Therefore, two-dimensional cuts through large, branched grains may let them appear as several individual grains within the same section. Kamb (1959) and Hooke (1969) have already discussed the statistical relevance of these branched grains. The sketches in Hooke (1980), Fig. 6 and more recently in Monz (2020), Fig. 3, further illustrate this issue that could result in over-represented clusters in the superimposed stereo plots from the different sub-samples.

*Fig. 6. The caption could be shortened by stating that the annotation is as in Fig. 5*

Changed.

*Line 192. The c-axis fabric has orthorhombic (and perhaps close to axial) symmetry, but is this also true of the stresses? What about the other two principal stresses. Are they consistent with plane strain or plane stress, as appears to be assumed in Fig. 9? Are the principal stresses and strain rates in this section of the glacier near the surface parallel to the flow direction ($\sigma 1$), vertical ($\sigma 3$) and horizontal ($\sigma 2$), with the lateral strain rate close to zero, as one would expect for a valley of constant width. One would expect the maximum principal stress to become inclined deeper into the ice as shear stress parallel to the base increases, which the modeled stress shows a tendency to do.*

We added the requested components to Figs 5 (eigenvectors) + 7 (stresses). The eigenvalues are named with $\lambda_1$ - $\lambda_3$ and the stress axes with $\sigma_1$ to $\sigma_3$ in decending order. The symbol size decreases respectively in both Figures.

We assume, that the eigenvectors are aligned with the strain rates (i.e. deformation). Due to the non-coaxial relation for simple shear, the eigenvectors could differ from the stress principal axes by up to 45°. Under this assumption,

the largest eigenvector in 79 m is perfectly aligned with the strain rate direction for dominating simple shear.

Figure 9 was removed due to speculative parts (see substancial changes).

*Line 200-204. This is a possible explanation, but I prefer the misorientation of the sample as the explanation, which as you state, fits very nicely when a 60o azimuthal 'correction' is made. A preserved fabric from earlier in the flow path is less likely at high temperatures when rapid recrystallization is expected.*

We removed this immature hypothesis, especially as we cannot see a smaller deviatoric stress in the uppermost parts.

*Lines 210-213. With this explanation you would expect σ1 to be vertical to explain the fabric at 79m depth and not as given by the numerical model. Although the vertical normal stress increases with depth, it is the deviatoric stress that controls deformation, not absolute stress values, and this likely does not change greatly with depth. I think the main thing that changes with depth is not the vertical effective compression (σ1 – σmean) but the increasing addition of base parallel shear stress, that in general terms increases linearly with depth.*

Our mistake was to consider a (hydrostatic) overburden pressure. However, this hydrostatic pressure does not contribute to the deviatoric stress that drives the c-axis orientation (via strain rates). We revised this part accordingly. As you say, the c-axis orientation (i.e. the centroid) in the deepest part is in alignment with shear stress: Base-parallel shear stress lead to a base-parallel orientation of the basal planes and thus a more vertical c-axis. The model shows that the shear stress component $\sigma^d_{xz}$ (which we will add to the results section) is the most dominant stress and the in-flow compressional component $\sigma^d_{xx}$ is much smaller in this depth (similar to $\sigma^d_{yy}$).

*Line 217. There is almost certainly some dependence of fabric on strain, which may not be great with fast recrystallization.*

We revised the details about recrystallization. Now, we consider, that strain-induced grain boundary migration with nucleation of new grains is the driving force. Then we do not have to assume any "fast" or "complete" recrystallization.

*Line 213-214. In simple shear the directions of principal stress are only aligned with those of principal strain for infinitesimal strains. The divergence grows as strain increases.*

We assume that strain rate (and the strain) and stress direction form an angle of ~45°. Therefore, the principal stress direction (governed by the simple shear component) in the deepest part of ~48° and the actual centroid (~2°) would fit under such an assumption. The MM cluster is aligned with the strain rate direction in that depth and not with stress as stress and strain diverge for simple shear.

*Line 238 I don't believe Cuffey and Paterson really explain why there should be*

*four maxima when the stress deviates from unconfined compression. This is more of an observation than an explanation.*

No, they only provide a description and mention different stresses are required for multi-maxima. In the revised version there is no need to cite them. We considered the more specific literature.

*Lines 244-245. This is unclear. A change in direction of glacier flow could be associated with either an increase or decrease in strain rate and thus decrease or increase in recrystallized grain size. Why just a decrease?*

This is correct, it could be an increase as well.
In our detailed description for SIBM-N and SIBM-O we discuss in detail, how the grain size evolves under different conditions.

*Line 251 and Table 2. Table 2 does not really give the grain size distribution, only average numbers of grains and average and maximum size in each sample. It would be useful to know the number of grains in each size category. Also interesting to know if there is any difference between the large and small grains in COF.*

We added a supplementary figure (Fig. 8, revised version) and median and 6 different grain size classes to Table 2 (now Table 4).

*Line 255-266. I'm not sure how much information is given by the air bubbles, except they do provide excellent evidence of active recrystallization by grain boundary migration. Bubbles are found both within grains and along grain boundaries both in temperate ice and in cold ice experiencing dynamic recrystallization, although the recrystallization mechanism may differ.*

As we found a better way to explain the recrystallization processes, we removed this Fig. 8

*Line 269. Hooke and Hudleston were concerned with polar, not temperate ice. The study was made on the Barnes Ice Cap.*

Thank you, again a valuable hint. We will change it:
They were observed in early studies on temperate glaciers (e.g., Rigsby, 1951; Kamb, 1959; Rigsby, 1960), ice capes with ice temperatures above -10°C (Hooke and Hudleston, 1980), and also in the bottom ice of Byrd Station and Cape Folger in Antarctica (Gow and Williamson, 1976; Thwaites et al., 1984). They are often referred as "diamond-shape" pattern or fabrics.

*Line 276. Whether the multimaxima fabrics are a result of unrepresentative sampling is still arguable in some circumstances, although the case you have here for these being true multimaxima fabrics is a strong one.*

Due to an additional analysis (Fig. 8 new manuscript), we can provide further arguments for the existence of multi-maxima pattern.

*Line 290-291. I think more data is needed to support this conclusion. The cores taken by Tison and Hubbard were in a different regime within the glacier – accumulation zone where perhaps there is longitudinal extension rather than compression, and close to the lateral margin of the glacier rather than in the center. This must lead to a more complex stress regime.*

Actually, some cores were drilled in the ablation zone and in these cores they found multi-maxima at the bottom. However, it is true that they are drilled at the margin and therefore they could significantly differ from our core close to the centre flow line of the glacier. This could complicate a direct comparison.

*Line 298-299. The combination of compression plus simple shear as applied in these experiments makes sense for much of your core, but not for the highest one where the shear component is minimal, nor for the lowest one, where the σ1 direction lies well outside the small-circle girdle of maxima. Some other explanation must hold in these places.*

Our revised interpretation assumes that the multi-maxima pattern clusters around the dominant strain rate direction, which is the one of base-parallel simple shear for this depth. The eigenvector is aligned with the expected strain rate direction (as you also describe). Due to recrystallization, we observe the clustering of four maxima around this axis.

*Line 300-301. I'm not sure if I'm properly interpreting what you are saying here, but the planes of maximum shearing stress in Duval's combined compression-simple shear experiments are not vertical and horizontal in his experiments, but inclined by an amount that depends on the relative amounts of normal compression and simple shear.*

This is exactly, what we wanted to cite here. His experiments show a multi-maximum pattern that is aligned with the compressional axis and two of these maxima are also aligned with the poles of the two shear planes as the angle between the maxima and the principal direction is roughly

*Line 312-316. Both Llorens et al. and Qi et a. are dealing only with simple shear, not with combined compression plus simple shear as in the torsion plus compression experiments. The conditions in the Rhone glacier I imagine change from horizontal compression with minimal base-parallel shear near the surface to horizontal compression combined with increasing base-parallel shear near the base of the glacier. As theory shows, shear stress increases approximately linearly with depth, while longitudinal stress stays approximately constant.*

This part has been removed. We cite Llorens to discuss potential localisation effects. However, Qi et al. are not providing any additional information useful for our interpretation.

*Fig. 9. The stress state shown in Fig. 9 is almost that of simple shear (no base-parallel longitudinal compression) with the shear plane (taken as the glacier bed) horizontal and σ1 inclined at 45o to the shear plane. If it is simple shear, there will be no horizontal compression and thus no shortening in the glacier flow*

*direction, which is incompatible with your data. If horizontal glacier flow-parallel compression is added σ1 will move closer to horizontal than it would be for simple shear alone. This looks like being the case for much of the glacier from the stresses shown in Fig. 7. I would expect the inclination of σ1 to be near zero at the surface and something less than 45o close to bedrock, the amount depending on the amount of horizontal compression. Although not a smooth change, the σ1 directions in Fig. 7 are consistent with this. The plot in Fig. 9 does not correspond to any of the plots in Fig. 7, all of which have σ1 at a shallower inclination than 45o and thus have associated planes of maximum shearing stress that are neither vertical or horizontal, unlike the situation in Fig. 9.
The one closest to horizontal thus cannot be considered a plane of simple shear. The 'shear plane' must always be the presumably sub-horizontal glacier bed.*

This Figure cannot hold a substantial revision. Therefore, we removed it and all parts in the text.

*Line 340. I disagree with the statement here (see comments for lines 210-213). Although the absolute value of the vertical normal stress increases with depth, the deviatoric vertical normal stress changes much less. It is the increase in base-parallel shear stress combined with the horizontal compressive stress (σxx if you like) that causes σ1 to rotate from near horizontal at the surface to inclined at some angle of less than 45° at the base.*

Indeed, the vertical stress $\sigma^{d}_{zz}$ is not responsible and actually decreases in our revised model with increasing depth. We will change this conclusion accordingly and agree with your suggestion that the orientation is driven by $\sigma^{d}_{xz}$ and $\sigma^{d}_{xx}$

*Line 342. The second part (ii) of the explanation for multimaxima fabrics given here makes no sense by itself. All states of stress that are non isotropic involve shear stresses. If the multimaxima fabric depended solely on the state of stress – that is with instantaneous adjustment of the c-axis fabric as the stress field changes – then there should be a constant relationship between the positions of the maxima and the principal stress directions. This clearly is not the case as the relationship in the deepest sample shows. There is, however, as you note, a consistent relationship between the fabric and the σ1 direction through most of the glacier and in all cases, with small deviations, the centroid of the COF fabric and the σ1 direction lie in the vertical plane that contains the flow direction. This is a key relationship that I believe you have only partly explained.*

We also put a larger emphasis on the fact that the general COF pattern is aligned with the flow direction (with an exception in 79 m).

**Substantial Changes:**

Introduction:
Here we flipped two paragraphs as it improves the readability of the manuscript. Furthermore we changed the quickly changing simple and pure shear interfering and changing compressional, extensional, and shear stress conditions along the valley in combination with

Data acquisition:
We moved the modelling part to an individual section and corrected some writing errors.

Ice flow modelling
We moved the details of the model to a new individual section.
We provide more details and additional basic equations (Glens flow law and Weertman's friction law) for a better understanding. Furthermore, we got some information about the basal velocities. We used these information to further constrain our model.
We also include details about sensitivity studies and explain more in detail why we use the respective parameters.

Crystal Orientation Fabric Analysis
We removed the details about the LASM measurements as they are no longer needed (see changes in Discussion).

Results
We added some more details about the eigenvectors (vaules and ranges) and a more precise distinction between mid-point and centroid. Furthermore, we considered literature recommendations from the reviewers.

Interpretation
Here, we added a general description about the stress conditions in the glacier and highlighted the dominant elements of the stress tensor. Furthermore, we calculate the strain rates and provide all modelled stress and strain rate components in tables. The strain rates cannot be derived directly from our model as we ran it in stationary fashion. Therefore, we calculated the strain rates via Glen's flow law.
We described more precisely the correlations between glacier flow and c-axes orientations for the observed azimuthal and co-latitudinal variations.
For this, we used the previously defined deviatoric stress components as references and to enhance the readability.

Especially in subsection 6.3 we removed the imprecise and speculative parts as criticised by both reviewers. Instead, we provide a detailed description about the recrystallization mechanisms that we believe are most important for our observations. Based on this RX-mechanism we provide an interpretation for the formation of the observed multi-maxima.
We also added further details to the results shown in Table 4 (previously Table 2). These classifications are more useful for our interpretation than the previously employed LASM scans. Therefore, we removed these LASM-scans and the information about them.

Discussion
In our discussion, we mainly revised the parts about recrystallization and restructured this section. Fat first, we compare our findings with other field studies, then with laboratory experiments and afterwards we describe in detail the recrystallisation. We explain in detail, why we follow the approach of Faria et al (2014) and explain the

difference to previous studies. Based on these findings we discuss the formation of the multi-maxima pattern.

We also consider additional hypotheses and mention them in the discussion. However, we removed the speculative part in conjunction with Fig. 9. This part cannot sustain in a thorough review.

Conclusion

Based on the changes in our discussion, we also rewrote parts of the conclusion. We mainly included the effects of SIBM-N and removed the (obvious and useless) statements that simple shear and compression are the main driving forces.

[revised manuscript text omitted]

---

## Referee Report (RR1)

Review of Revised MS "Crystallographic analysis ice on Rhonegletscher, Swiss Alps" by Hellman et al.

General

The paper has been considerably improved in this revised version, with the referees' comments largely addressed. It is good see some further details on the numerical modeling. It would be useful to add a brief discussion on the implications of using a model based on isotropic ice viscosity for natural ice with anisotropic fabric and thus anisotropic ice viscosity. The significance of this is implicit in the text in lines 206, 207.

There are still a few issues of interpretation of the COF in relation to the states of stress and strain, as I discuss below. But, with the uncertainties involved in knowledge of the states of stress, strain rate and strain in this glacier, there is room for a range of interpretations. Nonetheless, I do basically agree with the conclusions

Specific points keyed to line numbers in the text

Line 1. The centroid does not align with the modeled maximum compressive stress for the deepest sample. Perhaps this exception should be noted in the abstract, and the adverb "approximately" should be placed before "align."

Line 100. $\tau$ is the second invariant of the stress tensor, not $\tau^{n-1}$

Fig. 5. The labels $\lambda_1$, $\lambda_2$ and $\lambda_3$ should also be added to each of this projections, not just the first one.

Line 166. I really don't like the term "fissure" for these features, since it implies an opening, which is not there.

Line 216. $\tau_{xz}$ is the shear stress component, it does not represent simple shear unless the only deformation component in this reference frame is one of base parallel shear strain $\gamma_{xz}$.

Table 3. The borehole data on strain rate do not provide much useful information. Clearly the errors are large compared with the signal over the limited time between survey measurements. On a short time frame velocities can vary significantly (as the authors note), and this may explain why the model strain rates are poorly constrained by strain rates derived from the boreholes (especially the shear strain rates). The shear strain appears to be most strongly developed in the deepest ice, with less of a gradient in the top part of the profile than the model predicts. This may be due to the fabric related anisotropy and its effect on the flow law.

Line 235. Kinematics – the pattern of movement - is indirectly related to stresses, but in this case there is a close association. The word "causes" here may not be the best one, since a direct casual relationship is not established. "is associated with" might be better.

Fig. 7. The principal stress directions should be labeled $\sigma_1$, $\sigma_2$ $\sigma_3$ on these plots.

Line 240. Since this is a new paragraph, it's not clear which observation this refers to. If it's the relationship between fabric and $\sigma_1$ at 79m, this is not what Budd and Jacka (1989) show for multi maxima fabrics. The centroid of the fabric is vertical in both Budd and Jacka's examples and the ice at 79m, but $\sigma_1$ is vertical in Budd and Jacka, but at a strongly inclined angle to vertical for the ice at 79m.

Line 250. The ice c-axes tend to become oriented such that the basal planes are aligned for easy glide (have high Schmid factors), which may not be in the ice flow direction. It depends of course on $\sigma_1$, which in this case lies in the vertical plane following the flow. In the ablation zone, $\sigma_1$ would be close to vertical near the surface and the relationship between flow direction and crystals oriented for easy glide would be different.

Line 254. Again, this is not the "simple shear" component, just the shear component.

Line 260-263. This is not strictly simple shear, but it does approach simple shear towards the base. You might say "dominated by the shear component, which approaches simple shear." The last sentence of this paragraph is good.

Line 266-268. I may disagree here with what you appear to be saying. I believe the single maximum pattern seen in the deep parts of the Antarctic and Greenland ice cores is related to subhorizontal shear strain (close to simple shear) dominating the flow, as first suggested by Gow and Williamson. That is, they are related to shear strain, with the $\sigma_1$ direction inclined at some angle to the shear plane by an amount related to the degree of anisotropy of the fabric. I suspect the multimaxima fabrics with clusters arranged about a vertical line in these deep cores have a similar relationship to shear strain and stress that is rather like the situation in your deepest sample. Perhaps this is not in disagreement with you.

Fig. 8. A very nice addition!

Line 296. Not everyone thought that the multiple maxima were artifacts of limited sample size and sampling single grains several times, although this is something Monz et al. suggest may account for many of the early measured multimaxima fabrics. Kamb, in particular, was very careful to overcome this problem by the way he took samples.

I'm not sure what you mean here by "method-immanent."

Line 299-300. "High strain rate" is a relative term. Strain rates in valley glaciers are typically orders of magnitude smaller than those in most experiments. The key thing that appears to control whether or not multimaxima fabrics develop is temperature. Typical

multimaxima fabrics are restricted to "warm" ice, above about -10$^o$. As for strain rate, they appear to form under a wide range of strain rates. Russell Head and Budd (1979), for example, considered they might develop in nearly stagnant ice.

Line 305.  Kamb did not produce typical multimaxima fabrics in his 1972 experiments. They were either double maxima, in simple shear, or a small circle girdle in shear plus compression.  Also, Kamb noted that fabric development was mostly related to strain and only weakly to stress. Thus, Kamb may not be the best citation here to support your argument.

Peter Hudleston

---

## Author Response (AR2)

Dear Mr. Hudleston,

Thank you very much for your positive feedback. We considered your suggestions to further improve the manuscript. Please refer to our answers below for details.

Kind regards,

Sebastian Hellmann and the Co-authors

Regarding your comment about the model. This is actually a good point that needs to be clarified. We introduced a short discussion at the beginning of our interpretation:

We also calculate the strain rates $\varepsilon_{xz}$ from the stress tensor by using Glen's flow law, i.e. Eq. (1) (Table 3). Since the model does not consider anisotropy, the directions of stress and strain are parallel. This is a crucial limitation to be considered in the following interpretation. Especially for shear stress, the model-derived and the actual strain rate directions may differ significantly. In addition, the quantitative numbers deviate from isotropic ice as the aforementioned basal sliding affects the strain rates that an ice grain experiences. For more typical fabrics (e.g. single maximum, girdle fabric), enhancement factors could be introduced (Thorsteinsson et al. 2001, Pettit et al. 2007), but this is too complex for the multi-maxima patterns. However, we regard the modelling output as auxiliary values for our interpretation.

**Specific points keyed to line numbers in the text**

*Line 1. The centroid does not align with the modeled maximum compressive stress for the deepest sample. Perhaps this exception should be noted in the abstract, and the adverb "approximately" should be placed before "align."*

Thank you very much, we changed this accordingly.

*Line 100. $\tau$ is the second invariant of the stress tensor, not $\tau^{n-1}$*

Indeed the exponent should not be in the accompanying sentence. We corrected this error.

*Fig. 5. The labels $\lambda 1$, $\lambda 2$ and $\lambda 3$ should also be added to each of this projections, not just the first one.*

We adjusted the final figure.

*Line 166. I really don't like the term "fissure" for these features, since it implies an opening, which is not there.*

We removed the word here and only keep "fracture traces".

*Line 216. σxz is the shear stress component, it does not represent simple shear unless the only deformation component in this reference frame is one of base parallel shear strain σxz.*

We used the term simple shear here to distinguish it is not a pure shear. However, this is not fully correct and therefore we followed your recommendation and renamed it to shear strain.

*Table 3. The borehole data on strain rate do not provide much useful information. Clearly the errors are large compared with the signal over the limited time between survey measurements. On a short time frame velocities can vary significantly (as the authors note), and this may explain why the model strain rates are poorly constrained by strain rates derived from the boreholes (especially the shear strain rates). The shear strain appears to be most strongly developed in the deepest ice, with less of a gradient in the top part of the profile than the model predicts. This may be due to the fabric related anisotropy and its effect on the flow law.*

We agree that these components do not contribute much. Therefore, we followed your suggestions and removed them from Table 3 and the references in the following text.

*Line 235. Kinematics – the pattern of movement - is indirectly related to stresses, but in this case there is a close association. The word "causes" here may not be the best one, since a direct casual relationship is not established. "is associated with" might be better.*

We rephrased this sentence accordingly.

*Fig. 7. The principal stress directions should be labeled σ1, σ2, σ3 on these plots.*

We adjusted the final figure.

*Line 240. Since this is a new paragraph, it's not clear which observation this refers to. If it's the relationship between fabric and ʃ1 at 79m, this is not what Budd and Jacka (1989) show for multi maxima fabrics. The centroid of the fabric is vertical in both Budd and Jacka's examples and the ice at 79m, but ʃ1 is vertical in Budd and Jacka, but at a strongly inclined angle to vertical for the ice at 79m.*

This was a writing mistake. We considered the general alignment of the two azimuths (eigenvector and principal stress) and therefore Budd and Jacka (1989) are a good source to cite. We corrected the sentence:
The observed azimuthal alignment of the COF with the glacier flow (with limitations for 79 m) is in accordance with results from laboratory experiments […].

*Line 250. The ice c-axes tend to become oriented such that the basal planes are aligned for easy glide (have high Schmid factors), which may not be in the ice flow direction. It depends of course on ʃ1, which in this case lies in the vertical plane following the flow. In the ablation zone, ʃ1 would be close to vertical near the surface and the relationship between flow direction and crystals oriented for easy glide would be different.*

The statement should refer to our data and not c-axes in general. We rephrased the beginning of our sentence:
The ice crystal c-axes in our samples generally orient themselves parallel to the ice flow [...]

*Line 254. Again, this is not the "simple shear" component, just the shear component.*

We changed it to shear stress component.

*Line 260-263. This is not strictly simple shear, but it does approach simple shear towards the base. You might say "dominated by the shear component, which approaches simple shear." The last sentence of this paragraph is good.*

We considered your recommendation to enhance this paragraph:
This implies that the COF for the deepest sample is dominated by the shear component, which approaches simple shear.

*Line 266-268. I may disagree here with what you appear to be saying. I believe the single maximum pattern seen in the deep parts of the Antarctic and Greenland ice cores is related to subhorizontal shear strain (close to simple shear) dominating the flow, as first suggested by Gow and Williamson. That is, they are related to shear strain, with the $\hat{1}$ direction inclined at some angle to the shear plane by an amount related to the degree of anisotropy of the fabric. I suspect the multimaxima fabrics with clusters arranged about a vertical line in these deep cores have a similar relationship to shear strain and stress that is rather like the situation in your deepest sample. Perhaps this is not in disagreement with you.*

We agree with your description, but we do not want to focus on a particular sample (e.g. 79 m) but rather on the general multi-maxima clustering in all depths. The text does not make this fact clear. Therefore, we slightly rephrased it:

If the c-axis orientations would be governed solely by the orientation of the major principal stress and strain direction ($\sigma_1$) (mainly a result of compressional and simple shear stress), we would rather expect a single maximum in the stereo plots as in deeper parts of other ice cores (Faria et al., 2014a). As observed in Figures 5 and 7, there is no single maximum. Instead, the individual c-axes in our samples deviate on average about 30° from the principal stress or strain (for 79 m) direction (indicated by black small circle girdles in Fig. 7) and group in several maxima.

*Fig. 8. A very nice addition!*

Thank you. This was a recommendation of the second reviewer.

*Line 296. Not everyone thought that the multiple maxima were artifacts of limited sample size and sampling single grains several times, although this is something Monz et al. suggest may account for many of the early measured multimaxima fabrics. Kamb, in particular, was very careful to overcome this problem by the way he took samples. I'm not sure what you mean here by "method-immanent."*

Indeed, your paper Monz et al (2020) with the nice description of the branched grains encouraged us to be more strict. We added an "often debated to be". Method immanent: as (again shown in Monz) the thin sections cannot handle branched grains very well. However, we removed this detail as it is not relevant to discuss here.

*Line 299-300. "High strain rate" is a relative term. Strain rates in valley glaciers are typically orders of magnitude smaller than those in most experiments. The key thing that appears to control whether or not multimaxima fabrics develop is temperature. Typical multimaxima fabrics are restricted to "warm" ice, above about -10o. As for strain rate, they appear to form under a wide range of strain rates. Russell Head and Budd (1979), for example, considered they might develop in nearly stagnant ice.*

We adjusted this sentence:
The conditions for a "diamond-shape" pattern seem to be suitable in large glaciers like the Rhonegletscher with its high temperatures and large ice flow velocities compared to other valley glaciers.

However, as the Glacier Tsanfleuron in Tisson&Hubbard is also a temperate glacier, but with less obvoius MM-patterns, there need to be another difference than just temperature and we would like to point that out here. As you said in the first review, their glacier is much smaller and has a lower ice flow velocity. Therefore, we refer to this instead of "higher" strain rates.

*Line 305. Kamb did not produce typical multimaxima fabrics in his 1972 experiments. They were either double maxima, in simple shear, or a small circle girdle in shear plus compression. Also, Kamb noted that fabric development was mostly related to strain and only weakly to stress. Thus, Kamb may not be the best citation here to support your argument.*

According to the second reviewers recommendation, we shortened the discussion and in particular this paragraph was excluded.

Sebastian Hellmann and the Co-Authors

Dear Ms. Pettit,

Thank you very much for your critical feedback. We considered your suggestions to further improve the manuscript. Please refer to the point-to-point answers for details.

Kind regards,

Sebastian Hellmann and the Co-authors

**Specific points:**

*I still don't see a need for the 3D full stokes model when there is so little data to validate the model output and the model input and results aren't presented in the paper. Without seeing more of the model details and knowing that the model has been validated, I don't trust the subtle variations in stress presented in table 2 without some understanding of the uncertainties in the model. And the estimates of strain rate in table 3 are based on isotropic ice, so they are highly uncertain.*
*I would highly recommend starting with a simple estimate of stresses. If you would like some examples Pettit et al 2014 which uses a simple flow band model. Or Pettit and others 2007, which combines anisotropic effects with simple flow band. Or go back to Ny 1957 or the basic stress descriptions outlined in Cuffey and Paterson or Hooke or similar textbooks. It seems to me the overall surface flow pattern from the GPS is longitudinal compression along with weaker transverse extension (although without the numbers I can't estimate from the GPS). This should then be accompanied by vertical extension in order to conserve mass. The borehole shows very little shearing, which is expected since it is only the top 75% of the ice thickness. The fact that the maximum eigenvector gets more vertical with depth should be expected because the upper part of the ice column is dominated by normal stresses - with the longitudinal stress being the largest compressive stress and near the bottom, the shear stresses should increase, causing a rotation of the maximum eigenvector. None of that understanding requires the model. It would simplify the paper and minimize the tendency to believe details in a model that is not fully validated.*

We understand your point of view. However, we have to insist that the model was not set up from scratch only for our study. This model already existed, and we only used a subversion without time evolution and updated bedrock and ice thickness data (for 2017, the year of ice drilling). For a flow band model or any other model, our investigations in the vicinity of the drilling location are too uncertain (only a few weeks of observations) for setting up a completely new model.
Indeed, we could remove the model and just argue as you propose – surface data show that we have a compression and then increasing shear stress below 60 m from borehole investigations. However, we thought, it might be good to underlay these assumptions with a model available for this glacier. Furthermore, ElmerIce models are state of the art and were used for many glaciers in the last years (e.g. Aletschgletscher, Gauligletscher amongst others). Such a model (even the isotropic version) allows to investigate/consider the large scale ice flow effects that may have an influence on the microstructure.
As a last point, we did not only qualitatively argue, which stresses and strain rates are expected, but want to frame the argumentation/interpretation with quantitative (but auxiliary) values.

According to the second reviews comments, we added some limitations/concerns for such an isotropic model at the beginning of our interpretation section.

*The best result/conclusion is that the fabric indeed is pointing in the direction of the most compressive stress and it is not a simple single maximum - that observation alone suggests that recrystallization is happening. I really liked the thin sections showing the bubbles, as I thought that helped frame the story of migration recrystallization. Because these observations don't provide a direct cause-effect story for the multi-maximum - just an association - I am not sure that a page of explanation is necessary and the statement in the abstract claiming that the paper provides "an explanation" In fact the abstract is confusing, as it says there is significant shear stress - but there isn't significant shear stress until near the bed, below the level of borehole.*

We added the LASM image. We agree with your argumentation – It shows examples for migration recrystallization. However, we selected a slightly different part of the zoomed section showing some more distinct features related to SIBM-N.
We also reformulated the abstract and exchanged the "an explanation" by "provide indications and suggestions".
Indeed, the shear stress part is confusing and a leftover from our first version. This part is removed.

*The overall citations have been improved, I'd suggest including Pettit et al 2007 and Pettit et al 2011 in the section on how fabric relates to deformation history - there are many others that could be in there too, but the use of sonic measurements allows for a different perspective on the fabric deformation relationship.*

We added these papers to our introduction.

*Mostly the text has been improved for readability and typographical errors. I still see a few typos (capes instead of caps). The abstract still contains some general phrases that don't provide much information and I'd suggest making being more specific "key information" (which information?) "specific parameters" (which parameters?) "horizontal" should be longitudinal. "good agreement" still needs to be defined in a few places.*

We revised the abstract and rephrased these imprecise sentences. Furthermore, we added some numbers to explain the "good agreement" (variations of up to 26° for the colatitude angle ($0° <= \varphi <= 180°$)).

*To summarize - I'd suggest cutting out the model (adding in a simple stress description/analysis) and cut text from the long discussion at the end and then maybe add back in the images showing the bubbles and grain boundaries - which I thought was useful for the discussion of the migration recrystallization.*

We removed some less relevant details from the discussion to shorten it. In return, we added the LASM images again (although a slightly different part of the scan showing some more distinct features as the one in our first version). However, as said above, we would like to keep the model as it is also a powerful modelling code and followed the recommendations of Reviewer 1 for improving the modelling details although we also understand your concerns.

---

## Author Response (AR3)

Dear Olivier Gagliardini,

Thank you for these valuable comments.
Please refer to our point-to-point changes below.

Kind regards,

Sebastian Hellmann
on behalf of the co-authors

*- line 17: valuable insights on the \*\*current\*\* stress and strain distribution... I think it is more strain-rate than strain, as strain cumulates all strain-rate since snow deposit (which cannot be reconstructed from the current ice fabric as many different strain-rate history cumulating the same strain might give the same fabric in temperate ice). There are a number of places elsewhere in the manuscript where strain is used in place of strain-rate. Check this.*

Changed.

*- line 27: strains -> strain-rates*

Changed.

*- line 63: with studies from the last century -> with previous studies*

Changed.

*- Figure 3: Not so easy to visualise the compressive state with this plot. Why not plotting a map with the longitudinal strain-rate computed from the velocity field? (and also other components of the surface strain-rate using 3 subplots: Exx, Eyy and Exy). Explanation from lines 91-93 would be simplified.*

We combined this figure with Fig. 2 for better understanding and also changed the x-axis from "azimuth realtive to the ice core borehole" to "borehole index". Then, in combination with Fig. 2a and the ice flow therein it should be clear that the boreholes in the south are flowing slower than the ones in the north. We also adjusted the respective text.
In addition, we also include the requested strain rate plots and refer to them in an additional sentence.

*- line 105: was chosen with n=3 -> was chosen to n=3 (?)*

Changed to "was set to n=3"

*- line 108: what is the value of m?*

Added to the text: [...] while m = 3 and c are constant parameters. [...]

*- line 113: bedrock velocity -> sliding velocity*

Changed.

*- lines 129-130: you should mention first that fabric will induce an anisotropic viscous response of ice, especially because the viscous response is more anisotrope than the elastic one mentioned here (and is of better interest for glacier flow).*

We added this point: This results in an anisotropic viscous response of the glacier ice (Schulson and Duval, 2009, Chapt. 6).

*- line 160: I don't understand the z = y' and y = −z' notations. Is that true for all three types of thin sections? To help the reader, the two reference frames (x,y,z) and (x',y',z') should be plotted in Fig. 4.*

We adjusted our explanations and added the local coordinate systems (x',y',z') for the vertical sections and the reference frame (x,y,z). The horizontal section is already measured in the ice core reference frame (upper right corner of Fig. 4) which we now explicitly state in an additional sentence.

*- Figure 5: You should indicate the limit of the sample for c-axis orientation measurements. There are a number of very small grains at the limit of the sample that I guess are not measured (seems to be a result of the drilling technics which melt the ice)? Why not plotting the midpoint on this figure?*

We added a semi-transparent pale layer covering the area that we excluded from the analysis. The small grains around the ice core sample you mention arose from the liquid water that "glues" the sample to the sample holder and freezes immediately in the cold laboratories. This has nothing to do with the drilling and is a common feature in all FA images.
We added the midpoints to the stereoplots (as in Fig 7).
The minimum amount of pixels that is required to consider an area of similar pixels as "ice grain" is 500 pixels and already given in the text.

*- lines 218-219: No the Cauchy stress is not the deviatoric stress! And p is the isotropic pressure. So it should write: "model to derive the deviatoric stress tensor, which is defined as the Cauchy stress tensor minus the isotropic pressure, i.e.:"*

We removed this half sentence as it is explained correctly in the following sentence.

*- Table 2 would be more useful on the form of a graphic to visualize the evolution with depth of the stress components? Indicate also somewhere the sign convention that negative is compressive.*

We add a graph that shows the changes for each component with depth. Table 2 is obsolet now and therefore we removed it from the manuscript. However, we have to mention here that both reviewers asked for the actual numbers.

*- Caption of Table 3: I don't understand why you are using the borehole measurements to derive the strain rates? For a given depth, the ratio between a stress component and the corresponding strain-rate component should be constant (which seems the case). More explanations are needed on how the strain-rates are calculated.*

This "derived from borehole measurements" was a "leftover" from the previous version. We calculated the strain rates from the model and in addition from borehole measurements. The reviewers did not see benefits from the strain rates from borehole measurements. Therefore, we removed the values, but we have overseen this in the caption.

*- line 226: Not sure to understand what you mean by the direction of stress and strain-rate are parallel. More than parallel, using an isotropic law, you assume the collinearity of the two tensors (each components is proportional to the other S_ij = A D_ij with A a scalar)*
*- lines 228-229: The orientation dependent response of an anisotropic ice is certainly the main deviation, the alignement of the two tensors comes after. You can get up to an enhancement factor of 10 for anisotropic ice.*

For both comments, we changed the respective paragraph:
Since the model does not consider anisotropy, orientation dependent response of the anisotropic ice is not considered. For typical fabrics (e.g. single maximum, girdle fabric), enhancement factors could be introduced (e.g. Thorsteinsson, 2001; Pettit et al., 2007) to overcome this issue. Beyond, anisotropic flow laws (e.g. Gillet-Chaulet et al., 2005) can be employed to consider more complex fabrics such as the multi-maximum pattern. Furthermore, in isotropic models, stress and strain rate are connected with a scalar value and the principal axes of stress and strain rate tensors are parallel. These are a crucial limitation to be considered in the following interpretation. Especially for shear stress, the model-derived and the actual strain rate directions may differ significantly. In addition, the quantitative numbers deviate from isotropic ice as the aforementioned basal sliding affects the strain rates that an ice grain experiences. However, we regard the modelling output as auxiliary values for our interpretation.

*- line 231: I don't understand the "is too complex for the multi-maxima patterns". There are a number of micro-macro models that could (and have already) computed the anisotropic response of an ice sample for a given fabric. One could use for example the simple Static model assuming homogeneous stress in the crystals to evaluate the polycrystal response for such multi-maxima fabrics.*

Thank you very much for this remark. We rephrased this sentence: Beyond, anisotropic flow laws (e.g. Gillet-Chaulet et al. 2005) can be employed to consider more complex fabrics such as the multi-maximum pattern.

*- line 234: eigenvector of which tensor?*

Changed: eigenvectors of the stress tensors

*- line 244: Figs. 3 and 7*

"s" added and changed 3 -> 5

*- line 245: My understanding of recrystallisation fabrics is that they adapt almost "instantaneously" to the current state of stress. Not sure you need to mention the last four decades?*

In the first version we wrote "recently". However, this was said to be imprecise writing. Therefore we added a time frame. We remove the "in the last four decades" but keep the residual sentence as is.

*- line 271: strain or strain-rate?*

Changed: strain rate

*- line 272: I don't see easily why the presence of simple shear would induce that the principal directions are not anymore aligned, even for an anisotropic material (e.g. orthotropic)? May be add a reference here.*

In presence of simple shear stress there is a 45° "misorientation" between strain rate and stress. We added a reference that describes this.

*- lines 277, 280, 285 strain or strain-rate?*

Strain rate is correct in the first two and also the better term in the last case.

*- line 280: Why for the depth 79m, the strain (rate?) is used instead of the stress? And how do you get the principal direction?*

This is again the non-coaxial theorem for cases in which simple shear dominates. In such cases, the principal stress and principal strain rate axes are not aligned but tited from each other by 45°.

*- line 287: open bracket not closed.*

Added a ")" at the end of the sentence.

*- line 321: what do you mean by high temperatures? A temperate glacier is at the highest possible temperature ice can be?*

Changed: "these temperate conditions in Rhonegletscher are a prerequisite"

- lines 326-332: why not discussing this above when these processes are introduced.

We also followed your recommendation here. Please see the file for changes as several sentences were moved into the section 6.3.

---

## Author Response (AR4)

Dear Prof. Gagliardini,

Thank you again for your remarks. We answer to your concerns in the following sentences.
We revised Figs. 2, 3, 4 and 5.

*I think there is still an issue with the definition of the reference coordinate system you are using for expressing both the components of the stress and strain-rate tensors and plotting the fabric. It should be mentioned in a consistent way, where x, y and z are first introduced (i.e. for the strain-rate, line 96 of the tracked changes version) how x and y are defined (line 166 it is mentioned that z is pointing upward, should be mentioned vertical, also).*

We clarified this in the respective sentences. Please refer to the track changes file.

*But is x or y pointing to the geographical north should be specified. Also I am wondering if the results would not be more readable if x was defined as the main flow direction? At least, in term of stress and strain-rate, it would be easier to interpret.*

Since we prefer to have stereo plots and maps in a proper geographic system (North=up), we would like to keep the plots as they are, but we added a pictogram in Fig. 2,3 and 5 that indicates the x and y direction (155/65° N). Since the glacier changes its flow direction every few hundred metres from SW to SE and back to SW, such a geographic system seems to be better here, especially when comparing results with further work on that glacier.

*Moreover, line 232, in the discussion about the stress, \sigma_{xx} is referred as the longitudinal stress, which would suppose that x is aligned with the main flow direction (if not, then \sigma_{xx} is not the longitudinal stress).*

We checked this and we are sure that we always kept our reference system: x is defined as in-flow and y is the coordinate perpendicular to the flow. Therefore x is the longitudinal and y the transversal component as requested by Ms. Pettit.

*At the end, I am not sure that you have a consistent definition of the coordinate system all along the manuscript. If you are using different ones (which I think is not a good idea), then different notations should be used.*

Actually, when referring to the components, we should have a consistent coordinate system with x=longitudinal and y=transversal component and z=vertically upwards (Table 2). We double-checked this. Only in line 195/Table 1 we do not refer to the glacier flow system (x,y,z) but to geographic azimuths. However, here, we compare the glacier flow direction and the position of the clusters and need to use an independent reference frame. Otherwise could not find any issues anymore. However, we had to revise Fig. 4 and the respective text as we think this section was too difficult to understand. Now, we first rotate all thin sections to the global coordinate system (x,y,z) instead of only to the system of the horizontal thin sections and then add the azimuthal angle (155°) so that the plot fits onto a proper northwards pointing stereo net. Before we just bypassed this step by immediately rotating from the system of horizontal thin sections to N.

*In short, you should clearly define the reference frame (x,y,z) at the beginning of the paper and express all components (stress and strain-rates) in this reference frame, as well as plot the stereo plots in the same frame (instead of N-S plot if the reference frame is not linked to the North).*

Although we understand your point of view with the stereo plots, we prefer to keep N-oriented Schmidt nets, which is rather common in earth sciences. The pictograms added to the various plots clarify any potential confusions concerning the coordinate system.

*The color scale of Fig. 3 should better reflect sign changes (important for \epsilon_{yy}). In particular, is the sign change of \epsilon_{yy} aligned with the main flow direction should be highlighted, by either assuming a reference frame aligned with the main flow direction or by indicating on this plot the main flow direction.*

We adjusted the colour bar in a way that 0 is white, positive values are red and negative values are blue. An exception is $\epsilon_{xx}$ as it only contains negative values. Here we use a slightly different colour bar with cold colours to enhance the constrast. We also added the actual glacier flow coordinate system in the lower right corner of each image.
As this is a map view, we would like to keep geographic North pointing upwards.
Then, Fig. 2a, Fig. 3, and the stereo plots in Figs. 5, 6 and 8 are congruent with the map in Fig. 1.

Kind regards,

Sebastian Hellmann and the Co-Authors